# *Diplodia seriata* Isolated from Declining Olive Trees in Salento (Apulia, Italy): Pathogenicity Trials Give a Glimpse That It Is More Virulent to Drought-Stressed Olive Trees and in a Warmth-Conditioned Environment

**DOI:** 10.3390/plants13162245

**Published:** 2024-08-13

**Authors:** Giuliano Manetti, Angela Brunetti, Lorenzo Sciarroni, Valentina Lumia, Sara Bechini, Paolo Marangi, Massimo Reverberi, Marco Scortichini, Massimo Pilotti

**Affiliations:** 1Research Centre for Plant Protection and Certification (CREA-DC), Council for Agricultural Research and Economics (CREA), 00156 Rome, Italy; giuliano.manetti@crea.gov.it (G.M.); angela.brunetti@crea.gov.it (A.B.); lorenzo.sciarroni@crea.gov.it (L.S.); valentina.lumia@crea.gov.it (V.L.); sara.bechini@crea.gov.it (S.B.); 2Terranostra S.r.l.s., 72021 Francavilla Fontana, Italy; studioterranostra@gmail.com; 3Department of Environmental Biology, Sapienza University, 00165 Rome, Italy; massimo.reverberi@uniroma1.it; 4Research Centre for Olive, Fruit Trees and Citrus Crops (CREA-OFA), Council for Agricultural Research and Economics (CREA), 00134 Rome, Italy; marco.scortichini@crea.gov.it

**Keywords:** olive tree, Botryosphaeriaceae, *Diplodia seriata*, branch and twig dieback BTD, drought stress, heat stress, climate change

## Abstract

The fungi Botryosphaeriaceae are involved in olive declines in both the world hemispheres and in all continents where this species is cultivated. In Salento (Apulia, Italy), the Botryosphaeriaceae *Neofusicoccum mediterraneum* and *N. stellenboschiana* have been reported as the agents of a branch and twig dieback that overlaps with olive quick decline syndrome caused by *Xylella fastidiosa* subsp. *pauca*. In this study, we report the finding of *Diplodia seriata*, another Botryosphaeriaceae species, in Salento in *Xylella fastidiosa*-infected olive trees affected by symptoms of branch and twig dieback. Given that its presence was also reported in olive in the Americas and in Europe (Croatia) with different degrees of virulence, we were prompted to assess its role in the Apulian decline. We identified representative isolates based on morphological features and a multilocus phylogeny. In vitro tests showed that the optimum growth temperature of the isolates is around 25–30 °C, and that they are highly thermo-tolerant. In pathogenicity trials conducted over eleven months, *D. seriata* expressed a very low virulence. Nonetheless, when we imposed severe water stress before the inoculation, *D. seriata* significatively necrotized bark and wood in a time frame of 35 days. Moreover, the symptoms which resulted were much more severe in the trial performed in summer compared with that in autumn. In osmolyte-supplemented media with a water potential from −1 to −3 Mpa, the isolates increased or maintained their growth rate compared with non-supplemented media, and they also grew, albeit to a lesser extent, on media with a water potential as low as −7 Mpa. This suggests that olives with a low water potential, namely those subjected to drought, may offer a suitable environment for the fungus’ development. The analysis of the meteorological parameters, temperatures and rainfall, in Salento in the timeframe 1989–2023, showed that this area is subjected to a progressive increase of temperature and drought during the summer. Thus, overall, *D. seriata* has to be considered a contributor to the manifestation of branch and twig dieback of olive in Salento. Coherently with the spiral decline concept of trees, our results suggest that heat and drought act as predisposing/inciting factors facilitating *D. seriata* as a contributor. The fact that several adverse factors, biotic and abiotic, are simultaneously burdening olive trees in Salento offers a cue to discuss the possible complex nature of the olive decline in Salento.

## 1. Introduction

*Botryosphaeriaceae are an emerging/wide-spreading fungal taxon pathogenic to tree species*. Botryosphaeriaceae is a fungal taxon within Ascomycota representing a serious threat to several perennial species regardless the taxonomic group they belong [1]. Several fruit crops can be colonized and damaged by Botryosphaeriaceae such as grapevine [2], apple [3], citrus [4], nut fruits including almond, pistachio and walnut [5], tropical fruits such as avocado [6,7,8] and mango [9,10,11,12].

These fungal species cause bark canker and xylem discolouration in the root system, stem, branches and twigs, frequently progressing with a die-back mode and ultimately leading to canopy desiccation and plant death. Botryosphaeriaceae produce a number of pathogenicity factors such as cell wall degrading enzymes, small secreted proteins and phytotoxic secondary metabolites, which can contribute to symptom expression even in a temperature-dependent fashion [13,14,15,16,17,18]. These fungi can be saprobic on dead plant material, or they can colonize viable plants endophytically and revert to a pathogenic lifestyle when the host is weakened by stress factors such as drought and extreme temperature fluctuations [1,19]. This last point is becoming more and more crucial in view of the ongoing climate change which implies an exacerbation of drought and temperature increase. In these conditions, Botryosphaeriaceae seems to be an emerging threat [15]. However, Botryosphaeriaceae can be also primary pathogens as, for example, *Neofusicoccum parvum*, *Lasiodiplodia theobromae*, *N. mediterraneum* and *Neoscytalidium dimidiatum* [20,21,22,23]. Another important factor which strengthens the presence of this fungal taxon in woody plants is their host-neutral lifestyle, namely the compatibility with a wide host range. Equally relevant is the attitude of the different Botryosphaeriaceae species to collectively colonize the host plant, thus acting as a polyspecies. This means that a multiplicity of interactions occurs between the different species and even the host plant that, overall, contribute to disease onset and progression [24,25,26]. All these features depict a highly flexible lifestyle of Botryosphaeriaceae which facilitates the spread of these fungi, increases their dangerousness profile in wild plant communities, urban greenings and agricultural crops. Moreover, Botryosphaeriaceae species are distributed across all continents except Antarctica and are present in all terrestrial ecoregions with some species specifically adapted in temperate and Mediterranean-like ecosystems, and others endowed with a standing out capacity of adaptation ranging from the south to the north, also including the northern boreal forests [1,19]. 

*Botryosphaeriaceae and the olive tree*. For more than a decade, Botryosphaeriaceae have been reported in association with or as causal agents of severe olive declines characterized by twig wilting, branch desiccation, wood discolouration and bark cankers, possibly culminating in tree death. Moral et al. [27,28] reported *N. mediterraneum* as the most aggressive agent of such olive declines in California (USA) and Spain. Úrbez-Torres et al. [29] isolated several Botryosphaeriaceae species as well as other fungal taxa from declining olive trees in California. In the pathogenicity trials, *N. mediterraneum* and *Diplodia mutila* were identified as the most virulent species, thus being revealed as the main agents. Other Botryosphaeriaceae species such as *N. vitifusiforme*, *N. luteum*, *Botryosphaeria dothidea*, *L. theobromae*, *D. seriata* and *Dothiorella iberica* showed a low virulence when tested singularly [29]. In Turkey, the Botryosphaeriaceae species *Neoscytalidium dimidiatum* was reported as the agent of a severe and widespread olive decline [23].

A botryosphaeriaceous polyspecies was also found associated with an olive decline in Uruguay: in the pathogenicity test, *Neofusicoccum* species (*N. cryptoaustrale*, *N. luteum* and *N. occulatum*) resulted as the most virulent, *Diplodia* species (*D. seriata* and *D. mutila*) were the least virulent, and *Botryosphaeria* species (*B. wangensis* and *B. dothidea*) expressed an intermediate virulence [30]. 

In contrast to what had been reported in California and Uruguay, *D. seriata* was repeatedly reported in Croatia, as the main agent of olive dieback, being able to reproduce symptoms of bark necrosis and wood streaking, and to wilt artificially-infected two-year-old trees [31,32]. *Neofusicoccum parvum*, widely recognized as one of the most virulent Botryosphaeriaceae species whatever the host species, was also isolated from declining trees and expressed an evident virulence in the pathogenicity trials, though to a lesser degree than *D. seriata* [32].

*Botryosphaeriaceae are emerging in olive trees in Italy*. In Italy, several studies have linked Botryosphaeriaceae with dieback manifestations of olive trees — bark necrosis, wood discolouration, twig and branch desiccation. Specifically, *N. parvum* has emerged as a virulent agent on oleaster in the Sardinia region [33], on cultivated olive trees in the Veneto region (northern Italy) [34] and in the northern part of the Apulia region (southern Italy) [35]. Linaldeddu et al. [34] found also additional Botryosphaeriaceae species in several regions, Veneto, Lombardy, Sardinia and Calabria, including *D. seriata*; nevertheless, in the pathogenicity trials, this *D. seriata* isolate produced very small lesions four months after the inoculation, which is consistent with a very low level of virulence. *Diplodia seriata* was also isolated from declining olive trees in the northern part of the Apulia region by Carlucci et al. [35], but its pathogenicity was not studied. In Salento, the southern part of the Apulia region, Boscia et al. [36] reported in a review article the presence of *N. mediterraneum* in trees affected by Olive Quick Decline Syndrome (OQDS), a destructive disease widespread in that area and *tout court* attributed to *Xylella fastidiosa* subsp. *pauca* (Xfp) [37,38]. However, no details on the pathogenicity of this fungal species were reported. Recently, we isolated *N. mediterraneum* and *N. stellenboschiana* from olive trees in Salento [22,39]. Based on their pathogenic characteristics and degree of virulence, they were established as the agents of a branch and twig dieback (BTD) occurring in the olive plantings in that area and overlapping with OQDS. Symptoms of OQDS and BTD are confusingly similar as both diseases express leaf chlorosis, twig wilting and branch desiccation. However, the presence of irregularly scattered or wedge-shaped discolouration patterns in the wood was established as the main discriminating feature of BTD and Botryosphaeriaceae, whereas the apical leaf scorch was evaluated as a good phenotypic marker of OQDS and Xfp. Obviously, it is well understood that both symptoms are not strictly pathognomonic for these diseases [22,39,40]. 

While we are completing a survey throughout the Salento territory aimed at revealing the degree of overlap of Botryosphaeriaceae and Xfp, we are isolating additional Botryosphaeriaceae species. Among these and in addition to *N. mediterraneum* and *N. stellenboschiana*, a morphotype is emerging as widespread that was identified as *D. seriata*. 

Given the wide variations in virulence recorded for this species in the isolates from California, Uruguay and Italy (low virulence) [29,30,34] and Croatia (high virulence) [31,32], we considered it urgent to investigate the virulence of the Apulian isolate in order to understand its role within the Botryosphaeriaceae polyspecies infecting olive trees in Salento, and thus establish its possible contribution to BTD manifestation. 

In this work, we report the results of the morphological and multi-locus molecular identification of four isolates of *D. seriata* from declining olive trees in Salento [41]. Then, we present results of the in vitro growth in a wide range of temperatures, which is important to evaluate their adaptability to an environment increasingly subjected to severe climate changes. In vitro growth was also evaluated in relation to different degrees of water potential in the artificial medium, to establish any potential link with drought stress in the host plant. Important features of the pathogenicity and virulence were also revealed by performing: (i) trials suited to the canonical mono-factorial Koch postulate which assessed the virulence degree on non-stressed plants; (ii) trials suited on polyfactorial Koch postulate which enabled to evaluate the effect of water stress and the inoculation season on the virulence degree of *D. seriata*. An analysis of the meteorological data from Salento—temperature and rainfall—was performed over a period of the last 35 years to assess if conditions of thermal and drought stress are increasingly intensifying in the olive groves in Salento. 

In the discussion, we review the role of *D. seriata* in complex diseases involving important crop species as well as the abiotic factors and the host features which might condition the virulence of *D. seriata*. In the light of literature data and putting together all the results obtained in this work, we discuss the role of *D. seriata* in the olive decline occurring in Salento.

## 2. Results

### 2.1. Fungal Isolations and Detection of Xylella fastidiosa

Among the several botryosphaeriaceous isolates obtained from discoloured wood of BTD-affected olive trees, we selected four isolates in distinct Salento provinces (specified in Material and Methods)—CREA-DC TPR OL. 437, 458, 464, 700—as they appeared different from *N. mediterraneum* and *N. stellenboschiana* [22,39], but similar to each other. Real-Time PCR for detection of Xfp positively detected the target in all four BTD-affected trees from which they had been isolated.

### 2.2. Morphological Features and Cultural Characteristics of the Fungal Isolates

Colonies grown in axenic cultures on PDA were moderately fluffy and dirty white in the first three days of growth in the dark; soon after, the colonies assumed a brownish/olivaceous/greyish tonality with sparse or flat mycelium in the central area. Over time, all colonies assumed a stronger mouse grey-blackish or dark brown colour, with a dense mycelial mat and regular margins, except in OL.700, whose margins were uneven. The back side of mature colonies was blackish with a bluish shade (Figure 1).

After 10 days of culturing on pine needles deposed on water agar plates, isolates produced conidiomata pycnidial, partially immersed in the matrix, and emerging at maturity, blackish, solitary, globose and coated with hairy mycelium (Figure 2).

Conidia were mostly observed in the interior of the pycnidia, while exudating conidia were observed very rarely. Conidia lacked any persistent mucous sheath; they were initially hyaline, then becoming dark brown with a roughened surface, ellipsoidal, with one curved and one straighter side, apex rounded, base subtruncate. Septa were not observed. The mean size of conidia of the different isolates was globally in the range 24.9–26.8 µm (length), and 10.8–12.1 µm (width); length/width ratio ranged from 2.1 to 2.4. Extreme sizes were in the range 20.4–34.7 µm (length) and 9.1–16.1 µm (width). See Table 1 for the conidia size referring to each isolate and Figure 3 for the images. Overall, the conidia size and shape of Apulian isolates was like that reported for *D. seriata* by Phillips et al. [42] (size: 21.5–28 µm, 11–15.5 µm width), though Apulian isolates were somewhat longer.

We observed germination from 6 to 24 h after setting up the aqueous suspension. Germ tubes emerged from one or both apical ends or along the conidium side specifically in sub-apical position or in the middle of the side. Most germinating conidia showed no septa, but very rarely one/two septa were observed (Figure 4).

Growth of all isolates was recorded at 5–35 °C in the PDA test. The lowest growth rate was at 5 °C. With temperature rise, a progressive increase in growth was recorded, getting the highest value at both 25 and 30 °C (OL. 458 and 464), at 25 °C (OL.437) and at 30 °C (OL.700). A sharp decrease was recorded at 35 °C. In general, growth was significantly different among the different temperatures for each isolate; some significant differences were detected among the isolates at each temperature. In general, in the range of 15–35 °C, isolate OL.437 evidently showed the fastest growth rate. Growth was absent after an incubation at 40 °C for six days, but mycelium maintained the viability in all plugs. See Figure 5.

In the microtube-survival-test, viability was maintained in all the plugs of all isolates at 45 °C. At 50 and 55 °C, the percentage of viable plugs progressively decreased with temperature rise. Specifically, plugs of OL.700 and 437 were all devitalized at 55 °C. See Table 2 for details.

The growth rate of the *D. seriata*-like isolate CREA-DC TPR OL.437 at a natural daily temperature regime typical of summer 2022, was very similar to the optimal growth rate performed in incubator at 30 °C. In fact, it was somewhat lower than at 30 °C in the first trial and did not differ at all in the second trial (Figure 6). We also report the comparison with the growth rate of *N. mediterraneum* (CREA-DC TPR OL.427) and *N. stellenboschiana* (CREA-DC TPR OL.431) previously published in Manetti et al. [39] (Figure 6). This comparison shows that growth rate of the *D. seriata*-like isolate is much higher than these two additional Botryosphaeriaceae species.

Testing the growth rate of the isolates in conditions of low water potential in osmolyte-supplemented PDA, clearly showed that the isolates increased their growth rate in plates with a water potential of −1 and −2 Mpa compared with non-supplemented plates, though differences were not statistically significant. At −3 Mpa, growth rate turned out to be similar to that in control plates. Interestingly, in plates with a water potential from −4 to −7 Mpa fungal growth still occurred though at an increasingly slow rate (Figure 7). 

### 2.3. Sequencing and Phylogenetic Analysis for Species Identification

The ITS region and fragments of TEF1-alpha and TUB2 were sequenced in all four *D. seriata*-like isolates, CREA-DC TPR OL. 437, 464, 458, and 700. Multiple sequence alignments showed that the ITS region and TUB2 fragment were identical in all four isolates. Identity was found in TEF in the couples OL. 437–458 and OL. 464–700, which differed from each other for five single nucleotide polymorphisms (SNP) (98.98% nucleotide identity). Regarding the comparison with reference sequences of known *D. seriata* strains from the fungal biodiversity centre (CBS), nucleotide identity varied in the percentage range 99.59–100% (ITS), 93.33–100% (TEF1-alpha), and 92.56–100% (TUB2) (calculations made on the sequences trimmed to span the same region used for phylogenesis). Other Botryosphaeriaceae species had much lower values for nucleotide identity with the isolates under study.

In the phylogenetic tree, the relationships of *Diplodia* species fully matched the topology of the phylogeny of this genus reported by Yang et al. [43] and Zhang et al. [44]. Sequence data of the *D. seriata* representatives and the Italian isolates were included without uncertainty in the same clade with a high bootstrap value. To ensure best readability a reduced number of representatives of the genus were included in the phylogeny presented in Figure 8. A more complete phylogeny of the *Diplodia* genus is presented in Appendix A and fully confirms the results depicted in Figure 8.

Thus, overall, results of morphological observations and molecular analyses identified with certainty the fungal isolates from olive trees of Salento as *D. seriata*.

### 2.4. Pathogenicity Tests on Non-Stressed Olive Trees

With regard to the basic Koch postulate experiments, namely those performed on non-stressed olive trees cv Frantoio (stem and twigs) and Ogliarola (twigs), all trees inoculated with both *D. seriata* and sterile PDA developed strong healing reactions which had (almost) completely covered the native inoculation cut 11 months after the inoculation event. Thus, no evident necrosis affected the bark in all trees, Frantoio or Ogliarola, either fungus-inoculated or the controls. In the fungus-inoculated stems and twigs, a restricted discolouration—pale and threadlike—affected the wood and spread a few millimetres upwards and downwards of the wound inoculation. Specifically, in Frantoio, the wood discolouration average length (net of the inoculation wound) was 1.94 cm (SD = 1.29) in fungus-inoculated stems, and it was 0.69 (SD = 0.45) in the twigs. In Ogliarola, the wood discolouration average length of fungus-inoculated twigs was 0.14 cm (SD = 0.18). In the sterile PDA-inoculated stems and twigs of Frantoio and Ogliarola, the wood discolouration was also extremely reduced and ranged from 0.03 to 0.44 cm (net of the inoculation wound). 

The outcome of all inoculations on Frantoio is depicted in Figure 9 and Figure 10 and graphed with the column chart in Figure 11. Only in fungus-inoculated stems wood discolouration length was significantly higher than in PDA controls (*p* < 0.01) (Figure 11).

*Diplodia seriata* was reisolated with high frequency from all discolouration streaks of the fungus-inoculated plants but no fungi were isolated from sterile PDA-inoculated points. 

### 2.5. Pathogenicity Tests on Drought-Stressed Olive Trees

In the bifactorial pathogenicity tests, we designed the experiment considering drought as an inciting factor able to produce a drastic injury, as conceptualized by Manion [47]. We thus assumed that the sine qua non condition for the scientific reliability of the water-stress conditioned inoculation test was that the trees had to suffer a severe stress—leaf shedding—but without triggering an irreversible dieback desiccation (neither in short times nor in long times). Only in this way could any necrosis arising from the inoculation point be attributed solely to the inoculated fungus and not to a death process directly incited by the prolonged lack of water. Trees were inoculated when most leaf blades were rolled (Figure 12). 

Moreover, the canopies of the inoculated trees were monitored even after the removal of the fungus-inoculated twigs to check their viability throughout the months. Observations conducted at two and six months after the end of the water lack period (coinciding with the inoculation time) established that an extensive leaf desiccation and defoliation followed the appearance of the phenotype that marked the time of fungal inoculation, i.e., the leaf rolling (Figure 12). However, some viable and normal-looking leaves were still present, meaning that the leaf blade might unfold and return to the normal shape after restoration of irrigation if dehydration had not been excessive. In general, the bark layer of the twigs remained green-coloured, and twigs were able to resprout, thus showing that viability had not been compromised (Figure 12). 

In some twigs desiccation affected the distal ends, but still far away from the inoculation points. Just in two trees (1 and 8 of the summer trial) did all twigs result in being desiccated two months from the restoration of the irrigation, though the main stem remained viable. Overall, the outcome of the water stress observed over time confirmed that this experimental model was suitable to test the effect of fungal inoculation following a severe leaf-shedding-causing, but non-lethal, water stress. 

Recording symptoms in drought-stressed (DS) trees inoculated with *D. seriata* both in summer and in autumn revealed the presence of evident bark necroses and wood discolouration expanding basipetally and acropetally from the inoculation wound. Symptoms were much more severe in DS trees inoculated in summer compared with those inoculated in autumn (Figure 13, Figure 14 and Figure 15). 

In the DS trees inoculated in summer, bark necroses spread tangentially and completely girdled the inoculated twigs in eight trees out of fifteen. Thus, girdling index—the ratio between the tangential spread of the necrosis and the twig circumference—ranged from 0.4 to 1 with an average value of 0.8. In DS trees inoculated in autumn, some tangential spread also occurred with a girdling index ranging from 0.2 to 1 and an average value of 0.4. Necrosis also affected the xylem tissue behind the necrotized bark. In the summer trial necrosis had deepened transversally from one millimetre to the complete section, thus showing the modality of a wedge-shaped canker. In general, in the autumn trial, necrosis superficially affected the sapwood, except one case in which the complete transversal section was affected. 

In the WW trees inoculated with *D. seriata* both in summer and autumn, no leaf desiccation was observed on the canopy, an evident callus reaction had begun to overgrow the inoculation wounds at the time of recording and no necrosis was affecting the surrounding green bark tissue. A slight discolouration affected the wood just behind the bark tissue, advancing a few millimetres and without any transversal deepening (Figure 13, Figure 14 and Figure 15).

Specifically, relating the trees inoculated in summer, wood discolouration average length was 5.6 cm (SD = 1.76) in DS trees and 0.24 cm (SD = 0.19) in WW trees. Regarding the trees inoculated in autumn, wood discolouration average length was 2.3 cm (SD = 2.1) in DS trees and 0.06 cm (SD = 0.1) in WW trees. Therefore, drought stress caused an increase in the discolouration streaks of 23.3 and 38.3 times compared with normal watering in summer and autumn respectively. Bark necroses spanned wood discolouration but it was somewhat shorter. See Figure 15. 

In the sterile PDA-inoculated twigs, inoculation wounds were healing, and a reduced internal wood discolouration, net of the inoculation wound, was present and evidently higher in DS trees compared with WW trees (1.15–1.63 vs. 0.03–0.05), though much lower than infectious lesions in DS trees, but higher than infectious lesions of WW trees (Figure 15). This would mean that the drought stress condition would cause, per se, some wood discolouration, though more restricted than infectious discolouration streaks, treatment per treatment. 

*Diplodia seriata* was back-isolated with high frequency from all fungus-inoculated trees—DS and WW trees. Testing asymptomatic wood 3 cm above and below the ends of the discolouration streaks (DS trees) and inoculation wounds (WW trees), *D. seriata* was never isolated from WW trees, but it was isolated from five trees out of fifteen (33%) (summer trial) and from two trees out of fifteen (13%) (autumn trial) in DS trees, always below the lower end of the discolouration streak. This clearly suggests a preferential basipetal progression of the fungus in DS trees (a die-back trend). No fungi were isolated from any sterile PDA-inoculated points.

Figure 16 shows the daily minimum and maximum temperature trend occurring during the drought-stress-conditioned pathogenicity trials. 

### 2.6. Trends of Temperature and Rainfall over the Period 1989–2023 in Salento

We analysed the meteorological data (temperatures and rainfall) from Salento (Galatina and Mesagne), in a wide period, 1989–2023, to detect signals of climate change. Concerning the meteorological data of Galatina, the trend line of the average values of the temperatures referred to July and August, showed an increase of 1.12 °C and 1.76 °C for the maximum and minimum temperatures, respectively. The total amount of rainfall referring to July–August showed a decrease of 35 mm (Figure 17).

The same analysis, conducted with data from Mesagne, showed an increase of the average maximum and minimum temperature of 1.86 and 1.50 °C, respectively, while the amount of rainfall decreased of 18.5 mm (Figure 17). 

Interestingly, in the time frame 2007–2017/8, in Galatina and Mesagne, rainfall was constantly under the threshold of 42.2 and 51 mm respectively (except in 2012 in Mesagne in which rainfall was 87.4 mm) thus defining a wide time range of water shortage during the hottest period of the year (Figure 17). 

The trend of rainfall evaluated on the course of the entire year, in the analysed time frame (1989–2023), showed an increase of 20.44 and 53.96 mm in Galatina and Mesagne, respectively, and of 7.4 and 8.8 rainy days in Galatina and Mesagne, respectively. 

The number of days with maximum temperatures below or above 30 °C, referred to the period of July–August, was considered a good parameter to estimate any change in the heat stress occurring during summer. We calculated this parameter in Galatina. The trend line, relating the temperatures below 30 °C, showed a decrease of 5.8 days, while it showed an increase of 9.3 days for temperatures above 30 °C (Figure 18). To confirm the increasing temperature trend, when temperatures above 33.1 °C were considered, we recorded an increase of 3.3 days. 

## 3. Review and Discussion

*Diplodia seriata* is a cosmopolitan and plurivorous fungal species occurring on many plant genera and families. According to previous studies [1] *D. seriata* has been reported on 121 hosts by genus and globally distributed in 46 countries. Grapevine and apple are among the most important and the most studied cultivated crops worldwide. Infections by *D. seriata* and Botryosphaeriaceae on these host species are common and widely studied. Therefore, we explore these case studies to give an insight into the pathogenic style of *D. seriata*. 

### 3.1. Diplodia seriata and the “Botryosphaeria Dieback” of Grapevine 

*Diplodia seriata* is one of the most frequent Botryosphaeriaceae species occurring worldwide on grapevines affected by “Botryosphaeria dieback” (BD), a severe trunk disease characterized by bunch rot, bud necrosis, graft failure, lack of vegetative growth, shoot dieback, bark canker, wood discolouration, and eventually death of the plant. Indeed, *D. seriata* is part of the botryosphaeriaceous polyspecies, agent of BD. However, despite its strict association with BD, results of the pathogenicity tests conducted by different authors in different world countries are contrasting, as this species was reported as virulent, low-virulent or non-pathogenic [2,48].

Úrbez-Torres and Gubler [20] performed a study of the pathogenicity of the nine Botryospaeriaceae species—*B. dothidea*, *L. theobromae*, *D. seriata*, *D. mutila*, *N. parvum*, *N. luteum*, *N. australe*, *Do. viticola* and *Do. iberica*—isolated from BD-affected grapevine in California. They tested diverse isolates for each species in young and mature wood tissue as well as green shoots and in time spans covering very long observation periods, from 5 to 24 months. They showed there was a variability in virulence among the different *D. seriata* isolates as well as a differential pathogenic action by this species on the different tissue types. However, in their final assessment, *D. seriata* was ranked among the least virulent species, while *L. theobromae*, *N. luteum*, *N. parvum* and *N. australe* were the most virulent. In another pathogenicity study using potted vines, Elena et al. [49] compared 14 isolates of *D. seriata* obtained from BD-affected grapevines in Spain and showed that there was a variability in virulence. Nevertheless, it was concluded that *D. seriata* is a weak pathogen to grapevine because only small necrotic lesions in the wood were observed, and neither foliar symptoms nor bark canker formation around the inoculation wounds were observed during the experiments. Indeed, one year after the inoculation the most virulent *D. seriata* isolate caused a wood necrosis averaging the length of 2.5 cm, net of the inoculation wound. 

Throughout Australia, *D. seriata* has been assessed by far as the most prevalent Botryosphaeriaceae species in BD-affected grapevines [50,51,52]. Nine isolates of *D. seriata* (teleomorph: *B. obtusa*) obtained in a survey conducted in 16 vineyards in Western Australia were tested in a pathogenicity trial on detached cuttings of grapevines, but lesions did not extend beyond the inoculation point in a time span of five weeks, while *B. australis*, *B. rhodina* and *B. stevensii* expressed a relevant virulence depending on the isolate [50]. Pitt et al. [21] tested the pathogenicity of thirty-eight isolates of eight Botryosphaeriaceae species isolated from BD-affected grapevines—*B. dothidea*, *D mutila*, *D. seriata*, *Do. iberica*, *Do. viticola*, *L. theobromae*, *N. australe*, *N. parvum*—obtained in a survey in New South Wales and South Australia. Importantly, the experimental model was highly likely due to the fact that the pathogenicity was tested in the wood of the trunk of field-grown vines that were 15 years old. *D. seriata* was the least virulent, together with *Do. iberica* and *Do. Viticola,* and its virulence was estimated as weak/moderate, whereas *N. parvum*, along with *N. australe* and *L. theobromae,* produced the longest lesions. Qiu et al. [53] performed pathogenicity trials on detached dormant canes in a moist chamber, potted grapevines in a glasshouse, and 20-year-old grapevines in field conditions, and established that distinct *D. seriata* isolates from BD-affected grapevines in Australia had different degrees of virulence, but they were still less virulent than *N. parvum*, *L. theobromae* and *B. dothidea*. In this study, *D. seriata* was thus confirmed as a weak pathogen in grapevine [53].

In New Zealand vineyards, BD was associated with *N. australe*, *N. luteum*, *N. parvum*, *D. mutila* and *D. seriata* [54]. In pathogenicity trials on detached green shoots, *D. seriata* was the least virulent. Thus, it was not considered further for pathogenicity on green shoots or trunks of entire potted plants [54].

In Apulia (Italy), Carlucci et al. [55] tested, on grapevine, the pathogenicity of two isolates of *D. seriata* and eight additional Botrysphaeriaceae species—*D. mutila*, *D. corticola*, *Do. sarmentorum*, *Do. iberica*, *B. dothidea*, *N. parvum*, *L. theobromae* and *L. citricola*—all isolated from BD-affected grapevine in that area. It turned out that the isolates of *D. seriata* were the least virulent (together with *D. mutila*), whereas *L. citricola*, *L. theobromae* and *N. parvum* were the most virulent. This result was puzzling, as the isolation frequency of *D. seriata* from symptomatic vines was far higher than the other Botryosphaeriaceae species. Similarly, in Turkey, etiological investigation on BD-affected vines showed that *D. seriata* was weakly virulent, indeed the least virulent, when compared with the other Botryosphaeriaceae involved, *N. parvum*, *B. dothidea*, *L. theobromae* [56]. 

In a scenario in which pathogenicity trials underlined the low virulence of *D. seriata*, but given its wide spreading and distribution, Elena et al. [49] hypothesized that *D. seriata* may still cause serious infections on grapevines in natural conditions that are difficult to reproduce with artificial inoculations under greenhouse conditions.

To challenge the notion that *D. seriata* is a weak pathogen in grapevine, a report from Tunisia assessed *D. seriata* as the most virulent agent in BD-affected vines compared with *B. dothidea* and *N. luteum*, in a six-week trial performed on detached green shoots [57]. Recently, a survey in BD-affected vineyards of Northern Italy (Piedmont region) identified *N. parvum*, *D. seriata*, *D. mutila*, *B. dothidea* and other non-botryosphaeriaceous fungi. Pathogenicity testing on one-year-old potted cuttings over six months showed that *N. parvum* and *D. seriata* were the most virulent [58]. 

### 3.2. Diplodia seriata and the “Botryosphaeria Canker and Dieback” of Apple 

Similarly to grapevine, apple is also damaged by a botryosphaeriaceous polyspecies which causes the “Botryosphaeria canker and dieback” (hereinafter BCD) consisting of fruit rot, papyraceous and cracking bark canker, wood discolouration, leaf yellowing and dieback of the canopy. *D. seriata* is one of the most prevalent species involved in BCD. However, the virulence of *D. seriata* appears controversial and variable also in apple. In USA and Canada, declining and cankered apple trees were associated with *D. seriata* (teleomorph: *B. obtusa*), *B. dothidea* and *D. mutila* [59,60] but *D. seriata* resulted always in being the least virulent. Regarding the Canadian study, necrosis length averaged as low as 2.8 cm in *D. seriata*-inoculated trees compared with 5.9 cm in *D. mutila*-inoculated trees, six months after inoculation [60]. 

In Chile, BCD was associated with *N. arbuti*, *L. theobromae*, *D. mutila* and *D. seriata*. In the pathogenicity trials, *N. arbuti* resulted as the most virulent while the other species, *D. seriata* included, showed very slow progress of necrosis on the detached lignified dormant twigs tested in a time span of 3 months. Nevertheless, as in the cases described above, *D. seriata* was the most frequent species in BCD-affected apple trees, in the surveyed area of Chile [61]. In Uruguay *D. seriata*, *B. dothidea*, *N. parvum, N. australe*, *L. theobromae* and *D. mutila* were associated with BCD-affected apples. When tested in pathogenicity trials on twigs of entire apple trees, in a time span of 45 days, *N. parvum* and *N. australe* were the most virulent, while *D. seriata* was the least virulent and *B. dothidea* resulted as non-pathogenic. Once again, *D. seriata*—together with *B. dothidea*—was the most frequent in the surveyed areas [62]. 

In Northern Italy (Piedmont region) a study on a fungal trunk disease of apple characterized by symptoms traceable to BCD was recently carried out and revealed that Botryosphaeriaceae—*B. dothidea* and *D. seriata*—were the predominant fungal species. In the pathogenicity trials on entire plants, along a time span of four months, *B. dothidea* was the most aggressive compared with the other species tested. On the contrary, *D. seriata* was ranked as a secondary pathogen [3]. 

Additional results contradict that *D. seriata* is a weak pathogen in apple. In India, forty-eight *D. seriata* isolates from BCD-affected apples were compared in a pathogenicity trial on potted apple trees and always showed a substantial virulence, though variable among the different isolates regarding the incubation time before symptom expression and necrosis length. Of these, 10% were ranked as highly virulent, having caused necrosis on the stem longer than 15 cm [63]. Similarly, in South Africa, after a survey in one-year-old apple orchards and in apple nursery material affected by canker and dieback, 39 fungal species were isolated and tested in the pathogenicity trials. *D. seriata* was among the five most virulent species as it produced a necrosis averaging a length of 54 mm on branches of entire trees, five months after the inoculation [64]. In Argentina, *D. seriata* associated with BCD-affected mature apple trees, revealed a relevant virulence in the pathogenicity trials on entire trees and in a time span of three months. In fact, it produced necrotic lesions averaging a length of 48 mm, however always smaller than those caused by the other Botryospaeriaceae tested, *D. mutila* (65 mm) and *Do. omnivora* (73 mm) [65].

### 3.3. The Case of Olive Tree and Diplodia seriata 

In olive trees, similarly to grapevine and apple, *D. seriata* shows a wide variation in virulence depending on the isolates and the geographical world area, with a low virulence as a rule and a high virulence in a reduced number of cases. In fact, a low virulence was attributed to isolates from California, Uruguay and Italy [29,30,34], while the isolates from Croatia were ranked as highly virulent [31,32].

### 3.4. The Tunable/Ambiguous Nature of Diplodia seriata as a Pathogen

The apparently contrasting results from the pathosystems described above suggests that virulence of *D. seriata* can be largely tuned by several unknown or suspected factors. A relevant genetic variability was detected in the species, thus suggesting a variability in the genetic determinants of virulence [49,66]. In grapevine, Spagnolo et al. [67] showed that virulence of *D. seriata* varied depending on the plant phenological stage, being higher during flowering and veraison compared with the phase of separated clusters. *Diplodia seriata* also showed a diverse virulence on the different plant tissues of the stem, being higher on green shoots [20].

The fact that Botryosphaeriaceae species, including *D. seriata*, increase their aggressiveness when the host plants are subjected to abiotic stresses [19] further support the hypothesis that virulence of *D. seriata* has a changing/subtle nature and might reconcile apparently contrasting data regarding its strict association to disease symptoms and the low virulence expressed in mono factorial pathogenicity tests. Particularly, heat and drought stress, especially when combined, are considered key factors for the expression of grapevine trunk diseases (in which BD is comprised) due to their direct effects on the host, the pathogens, and the outcome of the host–pathogen interaction. Importantly, intensity, duration, and timing (i.e., the occurrence of the stress in pre- or post-infection) of the imposed stress are also crucial for the microbial virulence to be expressed [68]. In grapevine, Van Niekerke et al. [69] showed that several Botryosphaeriaceae species—*N. australe*, *N. parvum*, *L. theobromae* and *D. seriata*—were much more virulent in drought-stressed than in unstressed vines, with lesion length in the xylem and the pith declining linearly with increasing irrigation volume. Qiu et al. [53] found that a severe drought stress following inoculation with *B. dothidea*, *L. theobromae*, *N. parvum* and *D. seriata* caused noticeably longer necrotic lesions compared with non-water-stressed inoculated plants. Specifically, this was true for just one isolate of *D. seriata* out of the three tested, suggesting a consistent intraspecific variability also for the reaction to a stress condition. In the same study, incubation of the inoculated detached canes at 30 and 35 °C caused longer necrosis by all Botryosphaeriaceae, compared with incubation at 25 °C. Regarding *D. seriata,* one isolate caused the longest necrosis at both 30 and 35 °C, a second one at 30 °C and a third one resulted in insensitive respect to temperature variation, suggesting once again an intraspecific variability within this species [53]. Fernandez et al. [70] compared the virulence of *D. seriata* and *N. parvum* on well-irrigated vine plants and those subjected to drought and thermal stress. They showed that on normally irrigated vine plants, virulence parameters of *D. seriata* (length and area of discolouration) were just slightly higher than in non-infected controls; namely, the fungus caused a very slight alteration. On the contrary, in the treatments in which plants had been subjected to thermal and water stress the fungus resulted as more virulent, especially if both stresses were imposed together. Moreover, it seems that the impact of combined abiotic stress might increase the plant susceptibility to less virulent pathogens such as *D. seriata* rather than to *N. parvum,* whose virulence resulted as high with or without abiotic stresses.

Overall, the above reported revision of literature data suggests that both abiotic factors and genetic variability of some pathogenicity/virulence determinants might modulate the outcome of *D. seriata* colonization in the host plants. In the latter case, it is well known that a variety of pathogenicity factors are present in *D. seriata*, as well as in other Botryosphaeriaceae—multiple forms of small secreted proteins, cell wall degrading enzymes, a high number of genes involved in the production of highly diversified phytotoxic metabolites, enzymes able to metabolize phytoalexins [17].

### 3.5. Diplodia seriata Infecting the Olive Trees in Salento: The Case of This Study

In this study, we report the isolation and identification of *D. seriata* from declining BTD-affected and Xfp-infected olive trees in Salento. This study, as well as the previous two by Brunetti et al. [22] and Manetti et al. [39] is part of a step-by-step-investigation aimed at shedding light on the etiological contribution of the Botryosphariaceae polyspecies to the olive decline in Salento, in which BTD and OQDS result overlapped [41]. To date, our investigation on the spread of fungi and Xfp in the declining olive trees in those areas is still in progress. In this study we report the case of *D. seriata* being found to be associated with wilting of twigs and branches and wood discolouration, similar to previous findings regarding *N. mediterraneum* and *N. stellenboschiana* [22,39]. 

The in vitro characterization of the thermal requirements for mycelium growth of our *D. seriata* isolates clearly showed that the highest growth rate occurred at high temperatures, in the range 25–30 °C, and the isolates still grew at 5 and 35 °C. This was in line with data reported in literature for *D. seriata* for which the optimum growth temperatures comprised the range 23.29–30.8 °C and residual growth was still recorded at the extremes, 5 and 40 °C [3,58,68]. Importantly, the isolates kept their viability at much higher temperatures. Moreover, *D. seriata* showed a high growth capacity also at the natural temperature regimes occurring in the scorching summers of these recent years, even higher than as performed by *N. mediterraneum* and *N. stellenboschiana*. Overall, these findings draw attention to two important points: (i) the *D. seriata* isolates colonizing the declining olive trees of Salento, as well as those studied by other authors [68], are well adapted to a wide range of natural environments characterized by different temperature regimes, and this also explains why *D. seriata* is so widely distributed in the world [1]; (ii) their thermotolerance makes them especially competitive in hot geographical areas or those areas that are becoming hot due to the global warming. 

Results of the pathogenicity tests reveal that *D. seriata* from olive trees has also a tuneable, ambiguous, and changing nature as reviewed above for isolates of this fungal species from grapevine and apple. 

In the inoculation trials both with long and short duration, the non-stressed olive trees opposed a strongly resistant reaction to *D. seriata* at the bark level as the wound at the inoculation point completely healed in a fashion indistinguishable from that of the PDA-inoculated controls. In the wood, length of the discolouration streak was significantly higher compared with PDA-inoculated controls only in the fungus-inoculated stems. However, also in this case, the progression of the discolouration was restricted, indicating that the *D. seriata* isolate under study is very lowly virulent to olive tree. Though a strict comparison with pathogenicity trials by other authors is not possible due to the diversity of the experimental conditions, it seems that our isolates of *D. seriata* fall within the low virulent group which we have highlighted above, reviewing the case studies in grapevine and apple from all over the world. Regarding olive trees, the similarity with the low virulent isolates from California, Uruguay, and Italy (in regions other than Apulia) is also clear [29,30,34], rather than those high-virulent isolates from Croatia [31,32].

However, similarly to the experiments reported for grapevine, above cited, in which the contribution of drought and heat stress on the outcome of *D. seriata* inoculation was studied [53,69,70], the drought stress imposed to the olive plants prior to the inoculation worked as a strong predisposing/inciting factor for the virulence of *D. seriata* to be fully expressed. In fact, relevant necrotic lesions affected the bark and the wood of DS-fungus-inoculated plants just one month after the inoculation, compared with WW-fungus-inoculated plants and the PDA-inoculated controls. Interestingly, in the experiment performed in autumn, the average lesion length and the necrosis girdling capacity of DS-fungus-inoculated plants were reduced by 58.9% and 50%, respectively, compared to the corresponding treatment of the experiment performed in full summer. However, these lesions were still significantly more extensive than those obtained in the WW-fungus-inoculated plants which in turn were almost indistinguishable from wounds of PDA-inoculated controls. The temperature trend that occurred while carrying out the experiments clearly shows that it was very high during the summer trial and much lower in the autumn trial. Overall, results of the two experiments give a glimpse that the evident increase in virulence of *D. seriata* was due to drought stress, but the high summer temperatures contributed heavily to enhancing the effect of drought. This seems quite coherent with the study by Fernandez et al. [70] in which *D. seriata* was more virulent to grapevine when thermal and water stress were imposed to the host singularly, but especially when they were imposed together. 

Coherently with the virulence increase of *D. seriata* in conditions of drought stress, our isolates were able to grow in osmolyte-supplemented media even faster than in non-supplemented media, thus showing that *D. seriata* (and *N. mediterraneum*) is more active in an environment with low water potential (−1 and −2 Mpa) and keeps a residual growth capacity even at very low water potential (−7 Mpa). It is well known that water potential significatively lowers in olive trees subjected to drought stress to cope with a high evaporative atmospheric demand (high leaf-to-air vapor pressure deficit, D_l-a_) and low soil water (corresponding to a low water potential in the soil, Ψ_s_). Considering that water moves following a gradient in potential energy from where it is higher to where it is lower, this is why the olive tree decreases the water potential of its tissues under Ψ_s_ and indeed strong osmotic adjustment both in the roots and the leaves are key to realizing this [71,72,73]. It is well known that olive tissues can withstand very negative values of Ψ, as the wilting point for the olive ranges approximately from −2.5 Mpa to −3.5 Mpa, and a capacity to sustain values around −8 Mpa has also been reported [73]. Moreover, it has been hypothesized that a low tissue water potential may constrain the host cell metabolism, thereby preventing the production and translocation of carbohydrates and other secondary metabolites necessary for plant defence against biotic attack [74,75,76]. 

The findings obtained in this work suggest that *D. seriata* may impact the sanitary state of olive trees in the field where multiple stresses can occur simultaneously, especially heat and drought, which are the greatest causes for concern in ongoing climate change [72,73,77]. Interestingly, our results are consistent with the tree decline concept which states that several tree diseases are the result of predisposing, inciting and contributing factors interacting sequentially according to a spiral pattern [47,78,79]. Specifically, predisposing factors are those acting perennially and which operate a permanent stress. Though they do not cause an evident symptomatology, they facilitate the action of the inciting factors which have instead a limited duration, and can produce a drastic injury which weakens the tree to the point that contributing factors can overcome plant defences and cause further damage. In this model, global warming is typically considered as a predisposing factor, in addition to other factors. Drought is categorized as an inciting factor in addition to other factors (frost, insect defoliators) paving the way to contributor factors such as bark beetles and canker fungi. The process can even result in the death of trees. The pathosystem under study in this work can be closely attributed to this model, as the high summer temperatures can be considered a predisposing factor, as they did not cause a damage by themselves but facilitated the action of the drought that worked as an incitant as it caused a drastic injury—leaf shedding—and conditioned the host’s resistance to be completely overcome by *D. seriata*, which thus acted as the final contributor. It is worth mentioning that in the field and in a scenario in which also drought becomes quite constant but not with such an intensity to cause a complete leaf shedding, then it would act also as a predisposing factor together with heat. Thus, it is worth reflecting that these conditions tune according to *D. seriata* infections and virulence. In fact, in the field, drought action might variously bounce in the range of asymptomatic/symptomatic conditions (for instance leaf shedding affecting some portions of the canopy, etc.). Thus, in the spiral decline model, it is reasonable to frame drought as a predisposing or an inciting factor according to its intensity and duration. 

### 3.6. Climate Change Is Taking Place in Salento and Clearly Suggests That Rain-Fed Olive Trees Are Subjected to Drought and Heat Stress

Temperature and rainfall data of the last 35 years in Salento were analysed in this study to verify if heat and drought progressively exacerbated in the last decades. Interestingly, both the average maximum and minimum temperature in July–August increased (by 1.12–1.86 °C and 1.50–1.76 °C, respectively) and the occurrence rate of temperatures higher than 30 °C also increased. This temperature increase seems quite meaningful and worrying, especially when compared with the value of 1.59 °C, which is the global surface temperature increase over land above 1850–1900 in 2011–2020, estimated to be human-caused through emissions of greenhouse gases [80]. At the same time, though the yearly amount of rainfall increased, the amount of rainfall referring to July–August decreased, which is meaningful as this is the period in which the atmospheric water request in terms of evapotranspiration is the highest. Importantly, we detected an uninterrupted period of eleven years characterized by particularly scarce rainfall in summer, starting from 2007, and thus almost exactly coinciding with the first OQDS finds in Salento, which date back to 2008 [81]. This strict association strongly suggests a link between drought/heat and OQDS that can be explained with the spiral decline concept. Overall, a scenario of a progressive warming and of an increase of a summer-confined drought clearly emerges from our analyses. This reveals that rainfed olive groves in the Salento environment are being subjected to long times of increasingly intense thermal and drought stress. This might possibly push the species to overcome its critical eco-physiological thresholds [72,73]. It is worth mentioning that a precipitation value of 500 mm per year is considered the lower limit for commercial olive yields under rainfed conditions [82]. Importantly, the geographic distribution of the mean yearly rainfall below 900 mt for the period 1990–2000 in the Mediterranean Basin clearly shows that most areas in Northern Africa and several areas of Southern Europe were below this value even then [77].

Importantly, in Salento, not only does climate change increase drought, but ground-water exploitation also exacerbates the issue due to high demand from growing tourism and intensive agricultural activities. The depletion of ground water by these activities can slow down or impede the recharge of water storages in the aquifers when climate-driven drought decreases. Overall, this can bring the groundwater to tipping points which irremediably compromise its resilience, i.e., the capacity of water storages to recover [83]. 

### 3.7. Xylella fastidiosa subsp. Pauca Infections Impose, on Olive Trees, an Additional Condition of Drought Stress

Symptoms caused by Xfp in susceptible olive trees—leaf tip desiccation, leaf chlorosis, twig wilting and dieback desiccation of the canopy scaffold—clearly suggest that they originate from a condition of water lack from leaf and woody tissue, probably due to an unsustainable lowering of the water potential. In susceptible Ogliarola Salentina and Cellina di Nardò, investigations on the xylem vessels of the stem and petioles of infected plants evidenced several abnormalities: degradation of the pit membrane among adjacent vessels, and occlusion of the vessel lumen with tyloses, gums, pectin gels and bacterial aggregates. These alterations are potentially able to decrease stem hydraulic conductivity and cause cavitation; thus, they seem to be in a relation of cause and effect with drought stress and symptom manifestation. [84,85,86,87]. Results of transcriptomic analyses are also fully in line with this view as genes involved in the response to water stress are also involved in the susceptible/resistant response of olive to Xfp [88,89].

The presence of olive trees that remain asymptomatic in the Salento territory, though they have been Xfp-infected for a long time [90] and the fact that after Xfp infection a long incubation period is generally required for symptom expression, open a scenario in which olive trees suffer a condition of an increasing drought stress due both to the climate trend, depletion of water storage by human activities and latent/incubating Xfp infections.

Overall, it is thus conceivable that *D. seriata* might take advantage of this condition and thus contribute to the onset of olive decline.

### 3.8. Using a Morpho-Phenotypical Marker to Monitor Water Stress

In this study, we used a morpho-phenotypical marker to be sure that the drought stress had been perceived by the trees, namely the leaf rolling of 80–100% of the canopy. This was the signal for fungal inoculation followed by a progressive watering resumption. Successively, an extensive leaf shedding occurred to further confirm that trees had been severely stressed. The fact that twigs and branches resprouted gave the assurance that no necrosis had been directly caused by drought in the inoculation points. However, on the basis of these results, this study would pave the way to a characterization of the eco-physiological parameters which quantitatively define the predisposition status of olive tree to *D. seriata*. This would require measurements of stomatal conductance, water potentials of plant tissues at different levels and a leaf-to-air vapor pressure deficit, inter alia [72,73]. Moreover, Ogliarola Salentina and Cellina di Nardò should also be tested in the future in the inoculation trials, as they are the olive cultivars widespread in Salento and heavily affected by OQDS and BTD. 

## 4. Materials and Methods

### 4.1. Fungal Isolates

Previously, during a survey in an olive orchard located in the municipality of Mesagne (Brindisi province, Apulia, southern Italy), we found olive trees severely affected by BTD. Regarding this phytopathological case, we reported the presence of and characterized the pathogenicity of *N. mediterraneum* and *N. stellenboschiana*, Botryosphaeriaceae that turned out to be strongly involved in the disease [22,39]. In the present study, we focused on an additional botryosphaeriaceous isolate, different from the species cited above, which was also found to be associated with BTD in that survey (CREA-DC TPR OL. 437). We also included in the study three additional botryosphaeriaceous isolates which were revealed to be the same species as OL. 437. These isolates were collected from BTD-affected olive trees during an ongoing survey throughout the territory of Salento: CREA-DC TPR OL. 464 is from Nardò (Lecce province), CREA-DC TPR OL. 458 from Sava (Taranto province) and CREA-DC TPR OL.700 from Tricase (Lecce province). Pathogenicity trials were performed to characterize the virulence of the isolate OL. 437 and clarify any possible link with the natural symptoms. Sample collection and fungal isolations were as described in Brunetti et al. [22]. A concomitant detection of *Xylella fastidiosa* was also performed with the Real-Time PCR procedure by Harper et al. [91].

### 4.2. Morphological Features and Cultural Characteristics of the Fungal Isolates

Colonies grown on PDA plates (90 mm in diameter) were observed and photographed at different times to document age-related changes.

Conidiomata formation and sporulation were induced by culturing the fungal isolates on autoclaved *Pinus pinea* needles deposed on 2% water agar plates (WA) (Oxoid), incubated at 23 °C under near-UV light [92]. Fungal structures were observed under a light microscope—Leica DM6B, equipped with Leica LAS X software, version 3.4.2, to record magnified depictions, and a Leica DFC 7000T camera for image acquisition (Leica Microsystems Srl, Milan, Italy). Conidia germination was observed in crushed pycnidia. 

To determine the optimal growth temperature and the cardinal temperatures for the viability of all four isolates under study, the growth rate and viability were assessed at temperatures ranging from 5 to 55 °C with a step size of five. 

In the range 5–40 °C, the test was performed on PDA plates in the dark; for each temperature, five PDA plates, 90 mm in diameter, were inoculated with a fungal plug 9 mm in diameter taken from an actively growing colony. The diameter of each colony was measured twice at right angles after the colony had covered roughly 70% of the surface for the fast-growing rates (from 15 to 35 °C), or after 15 days at 5 °C and 7–10 days at 10 °C. The daily growth was then calculated dividing the final average growth value by the number of days. When no growth occurred (40 °C), incubation lasted six days, then the plates were transferred at 25 °C to verify fungal viability. 

In the range 45–55 °C, the incubation was aimed at evaluating the fungus’ survival. As we had previously assessed that mature mycelium of Botryosphaeriaceae is more resistant to high growth-inhibiting temperatures [39], we sampled mycelial plugs from twenty-day-old colonies. At each temperature and for each fungal isolate, fifteen mycelium plugs were tested in five 2-mL tubes (three plugs for each tube) by incubation in thermal block for six hours. Then, survival was verified by transferring the plugs in PDA plates at 25 °C.

To assess the growth rate of the botryosphaeriaceous Apulian isolate CREA-DC TPR OL.437 in the regime of high summer temperatures occurring during 2022, we compared its growth rate at 30 °C in an incubator with that obtained by exposing the PDA cultures to the natural daily temperature variation in the external environment. Outdoor cultures were protected from direct sunlight by keeping them under a portico with natural shadow conditions. All cultures were covered with an aluminium foil. The test design and the recording were as described above for the determination of the growth rate under controlled conditions. The temperature of the outdoor test was monitored on a 30 min basis throughout the duration of the test using a data logger (Cryopak Escort iMINI Temperature Data Logger, Sydney, Australia).

With the aim of better understanding the behaviour of the four botryosphaeriaceous isolates in drought-stressed olive trees, their in vitro growth ability was assessed under different levels of water potential (Ψ) in a potato dextrose-containing medium (24 g potato dextrose broth plus 20 g technical agar, per litre). For comparison, an isolate of *N. mediterraneum* (OL.427) was also included in the study. Media were prepared as described by Aujla and Paulitz [93], according to whom increasing amounts of KCl were added to the medium to obtain ψ from 0 to −7 Mpa (specifically, see Table 4 of the cited article). For each isolate and ψ value, five PDA plates were inoculated as described above, and incubated at 25 °C for 48 h (fungal isolates under study) and 72 h (*N. mediterraneum*). Final growth recording was as described above for the determination of the growth rate under controlled conditions.

### 4.3. Sequencing and Phylogenetic Analysis for Species Identification

The botryosphaeriaceous isolates were subjected to multi-locus sequencing. After extracting fungal genomic DNA (gDNA) from axenic fungal cultures, the complete ITS region, fragments of beta-tubulin 2 (TUB2) and translation elongation factor 1-alpha (TEF1-α) were amplified and sequenced. Sequences were used as queries to find matches in the NCBI GenBank; they were aligned with known *D. seriata* isolates from CBS, and eventually used to infer a multi-locus phylogeny for species determination. 

All details on fungal gDNA extraction, primers, reaction assembly and thermal cycling of the amplifications are reported in Appendix A and references therein [94,95,96,97]. PCR amplicons were directly sequenced in both directions by Sanger technology (Bio-Fab Research s.r.l., Rome, Italy). Sequences newly generated in this study were deposited in the NCBI GenBank with Acc Nos PP712106-9 (ITS) and PP727264-71 (TUB2 and TEF1-alpha), and they were also included in Appendix A. They were compared with GenBank accessions using the blastn suite on the NCBI server (https://blast.ncbi.nlm.nih.gov/Blast.cgi?LINK_LOC=blasthome&PAGE_TYPE=BlastSearch&PROGRAM=blastn) (accessed on 10 April 2024).

Based on the blast results, a *Diplodia*-specific phylogeny was inferred to confirm the specific nature of the isolates under study. A combined multiloci dataset was used as input for the analysis: ITS + TEF1-alpha + TUB2. The reference sequences were extracted from the NCBI GenBank. Phylogenies in the taxonomic study by Zhang et al. [44] were considered as models to verify the reliability of our analyses. Generally, we included in the analysis reference species and strains with all three sequence loci, except one (CBS 140350 with ITS and TEF1-α only). *Lasiodiplodia theobromae* was included as the outgroup. The complete list and details of the reference sequences used in the phylogenetic analyses are presented in Spreadsheet S1.

Sequences of the isolates under study were aligned to reference sequences, each locus separately, with MAFFT on the EMBL-EBI server (htpps://www.ebi.ac.uk) (last access on 7 March 2024). The alignments were then trimmed to span the same region and concatenated in MEGA X [46]. 

The evolutionary history was inferred by using the Maximum Likelihood (ML) method and a general time-reversible model [45]. Bootstrap values were calculated from 1000 replicates. Initial tree(s) for the heuristic search were obtained automatically by applying Neighbor-Join and BioNJ algorithms to a matrix of pairwise distances estimated using the Maximum Composite Likelihood (MCL) approach, and then selecting the topology with a superior log likelihood value. A discrete Gamma distribution was used to model evolutionary rate differences among sites. Evolutionary analyses were conducted in MEGA X [46].

### 4.4. Pathogenicity Tests—Mono and Bi-Factorial Koch Postulate Experiments

The inoculation trials were carried out to assess the pathogenicity of the botryospaeriaceous isolate CREA-DC OL.437. Olive trees, cv. Frantoio, were supplied by Spoolivi-Società Pesciatina d’Olivicoltura (Pescia, PT, Tuscany, Italy). Olive trees, cv. Ogliarola Salentina, were supplied by Vivai De Nicolo Piante Mediterranee da Ricoltivare (Terlizzi, Ba, Apulia, Italy). The pot substrate was Radicom (Vigorplant, Lodi, Italy), which contained a mixture of peat moss, black peat, marsh peat, and vegetable compost. Humus was present in the form of humic and fulvic acids with a water pH of 6–6.5. Two-year-old plants were transplanted into 23 L pots. Older plants were kept in 30 L pots.

Basic Koch postulate experiments: we inoculated the basal portion of the stem at a height of 15–20 cm (average diameter 1.2 cm) and, in different trees, one-year-old twigs (average diameter 0.6 cm). For each trial, we used 10 plant replicates for fungal inoculation and four for inoculation with sterile PDA (the negative control). Regarding the twig trial, we inoculated two twigs for each plant replicate both in the fungal and control treatments, thus providing 20 replicates for the fungal-inoculated plot and 8 for the control. Cv Frantoio trees at the age of two and a half years old were used for inoculation on the stem and the twigs. Seven-year-old olive trees, cv Ogliarola salentina, were used for inoculation on the twigs. Plants were monitored for eleven months before performing destructive recording. 

Drought stress-conditioned Koch postulate experiments: in these experiments we used trees, cv Frantoio, at the age of two and a half years old and tested the effect of severe drought stress on the outcome of the subsequent fungal inoculation. Water stress was induced by suspending irrigation till the appearance of a stress phenotype on the canopy of the trees consisting in leaf rolling in 80–100% of the canopy. At this stage fungal inoculation was performed, and soon after watering was restored by supplying 300 mL water three times a week in the first week, 500 mL in the second and third week, and then more abundantly to maintain the soil substrate at field capacity. In each trial fifteen drought-stressed (DS) and as many well-watered (WW) trees (the non-stressed control) were inoculated on one-year-old twigs (average diameter 0.7 cm). Two twigs were inoculated for each tree, one with the fungal species under study, the other with sterile PDA which served as control. In the first inoculation trial, all trees, DS and WW, were inoculated on 24 August (summer inoculation). In the second trial DS and WW trees were inoculated in October (autumn inoculation). In this trial, inoculation was performed staggered in small groups containing an equal number of DS and WW trees that reached, each time simultaneously, the same stress phenotype (Figure 16). All plants were monitored for 35 days before performing destructive recording.

All inoculations—both basic and drought stress-conditioned—were performed as follows: rectangular PDA-mycelium plugs (14–16 × 3–4 mm sized for twig, and 18–20 × 8–10 mm for stem inoculation) were cut from an actively growing colony and placed on similarly sized wounds, i.e., on the xylem surface, which had been exposed by cutting the bark top-down with a razor blade through the cambium. The bark strip was then gently set on the plug. The inoculation point was covered with a sterile cotton disk that had been wetted with 3.5 mL sterile water and wrapped with a sterile strip of aluminium foil, which was taped to the stem at the top and bottom edges. The cover was removed after 30 days. 

In Table 3 we report details of all mono and bi-factorial pathogenicity trials performed from 2022 to 2023. 

At the end of the trials, the lengths of the external bark reactions and internal wood discolourations were recorded in the fungus-inoculated trees. The girdling index was also calculated as the ratio between the tangential spread of bark necrosis and the stem circumference. The length of the external healing reactions and of the corresponding wood discolouration behind them was also recorded in the stems/twigs inoculated with sterile PDA. To perform fungal re-isolation, eighteen wood fragments were collected from discolouration streaks in the fungus and sterile PDA-inoculated trees and plated on PDA dishes supplemented with 0.3 g/Lt streptomycin.

Regarding water stress-conditioned Koch postulate experiments, the final recording implied that, apart from the fungal re-isolation, the whole plants were maintained in the nursery for additional six months to monitor the progress of the stress phenotype induced by drought stress in terms of defoliation and a possible desiccation or resprouting.

### 4.5. Analysis of Meteorological Data from Salento

Meteorological data, in terms of rainfall (mm) and temperature (°C), were recorded over a period of 35 years, from 1989 to 2023, in two different sites located in Salento, Galatina (LE) and Mesagne (BR). Calculations referred to (i) a two-month period—July and August—which is representative of the hottest period of the year in the Mediterranean areas, and (ii) the whole year. Graphics with bars and trend lines were inferred in Microsoft Excel. The equation of the trend lines enabled the calculation of the increase/decrease of the meteorological parameters over the analysed period. Specifically, we used the daily maximum and minimum temperatures to calculate the average value of maximum and minimum temperatures for the two-month-period, in each year. Regarding the rainfall, we calculated the total amount in mm and the number of days of rain, either over the two-month period or the whole year. Moreover, regarding the site of Galatina only and referring to the period of July–August of each year, we calculated the number of days in which the maximum temperatures were below and above 30 °C, considering that this too might be a useful parameter to evaluate an increase of a potential thermal stress for plants. 

Raw temperature data were downloaded from the historical archive of ilMeteo.it (https://www.ilmeteo.it/portale/archivio-meteo/Galatina—accessed on 5 March 2024) (Galatina) and from the historical archive of the Protezione Civile (https://protezionecivile.puglia.it/annali-idrologici-parte-i-dati-storici—accessed on 8 March 2024) (Mesagne). Raw precipitation data were entirely downloaded from the historical archive of the Protezione Civile (https://protezionecivile.puglia.it/annali-idrologici-parte-i-dati-storici—accessed on 12 March 2024).

### 4.6. Statistical Analyses

A one-way, fully randomized analysis of variance and the Tukey test as post-hoc analysis were carried out to compare the fungal growth at different temperatures or on media with different ψ, and the length of wood discolouration streaks and bark cankers in the inoculated trees. PAST version 4.10 was used for the analysis [98].

## 5. Conclusions

To dispel any doubts: our research approach to the olive decline in Salento does not aim to deny, neither explicitly nor implicitly, the crucial role of Xfp that has been previously assessed [38,87]. Instead, it is aimed at shedding light on important additional factors possibly acting in a spiral-shaped decline. Roles of Botryosphaeriaceae have been previously demonstrated [22,39] and the possible interactions with abiotic factors and Xfp have been discussed [40,99]. In this paper we add another piece to the jigsaw as we clearly show that drought and heat negatively tune the resistance of olive trees to *D. seriata*, a Botryosphaeriaceae species which is emerging and widely spread in decline-affected olive groves in Salento [41]. It is thus conceivable that *D. seriata* might have a role in a spiral-shaped decline in which Xfp and additional Botryosphaeriaceae—*N. mediterraneum* and *N. stellenboschiana*—as well as climate-, antropic activity- and Xfp-driven drought/heat stress are involved. 

The awareness that fungi, in addition to Xfp, contribute to damaging the olive trees in Salento would be key to understanding which sanitary measures are more effective in reducing the expression of symptomatology and its spread [100,101].

To give strength to the hypothesis that the etiology of the olive decline in Salento has a complex nature are the data obtained from monitoring surveys in the containment and buffer zones of the advancing edge of the disease. The analysis of these data showed that in most of the sampled OQDS-affected olive trees, the bacterium was not detected. For example, during the monitoring surveys of 2021, the percentage of olive trees with clear symptoms of OQDS that tested positive for Xfp was merely 3.21% [90].

Currently, a survey aimed at assessing the degree of overlap between Xfp and fungi in the declining olive trees is nearing completion. Importantly, additional multifactorial Koch postulate experiments exploring the connections among the various actors in play are also in progress [41]. This is what is needed to clarify the matter.

## Figures and Tables

**Figure 1 plants-13-02245-f001:**
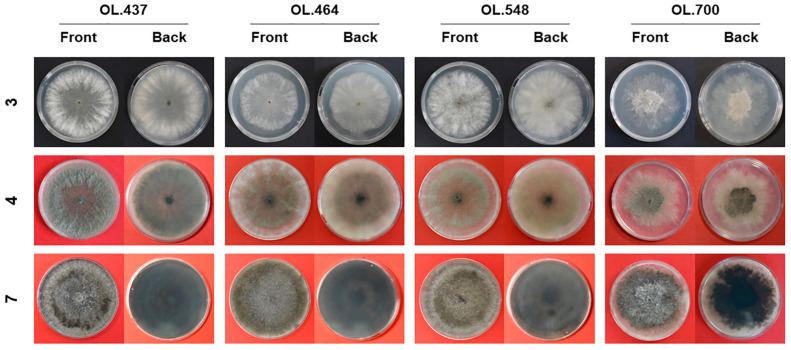
Axenic cultures of the botryosphaeriaceous isolates under study, on PDA (CREA-DC TPR OL.437, 464, 548 and 700). On the left, age of the cultures in days.

**Figure 2 plants-13-02245-f002:**
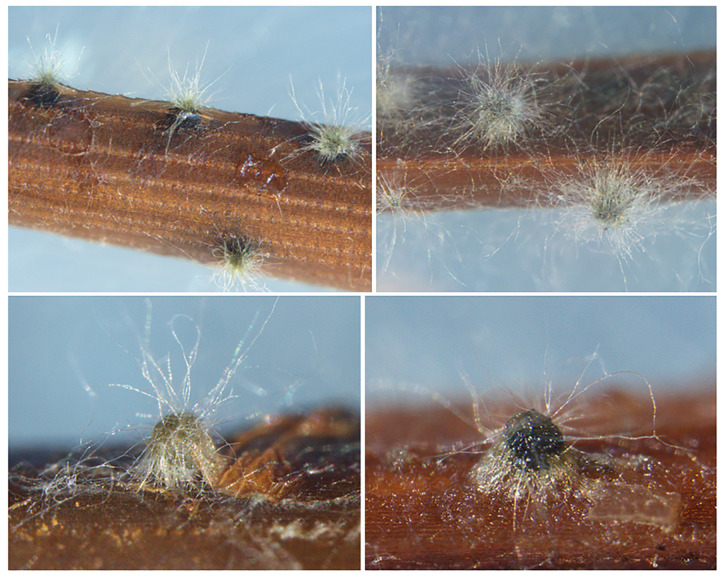
Conidiomata of the botryosphaeriaceous isolates under study, coated with hairy mycelium (developed on pine needles in water agar plates).

**Figure 3 plants-13-02245-f003:**
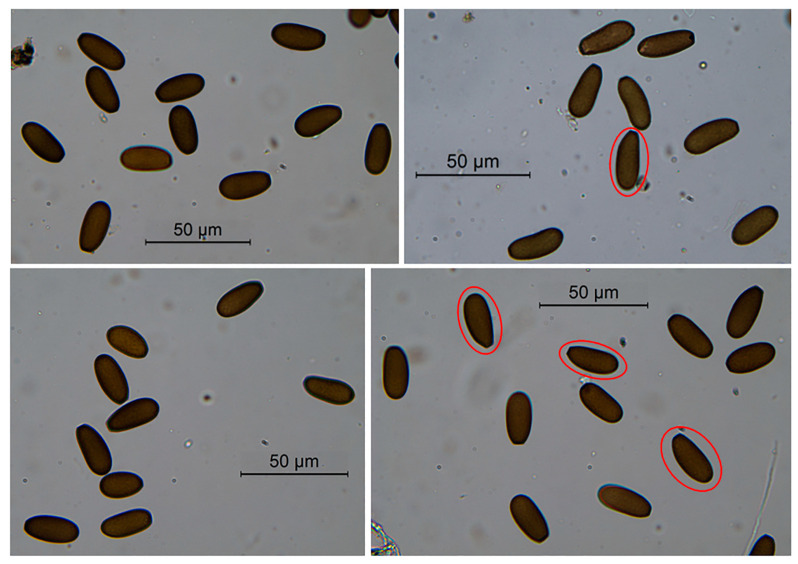
Conidia of the *Diplodia seriata*-like isolates under study. Conidia were observed after squashing the conidiomata which had been developed on pine needles; mucoid exudates were not observed; conidia circled in red clearly show that one side is more curved, and the opposite side is straighter.

**Figure 4 plants-13-02245-f004:**
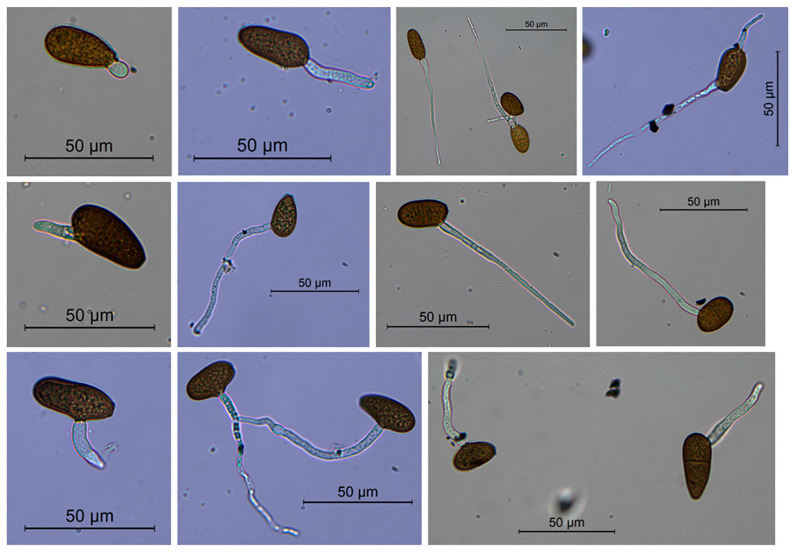
Conidia of the *Diplodia seriata*-like isolates under study. Germinating conidia with germ tubes arising from different positions.

**Figure 5 plants-13-02245-f005:**
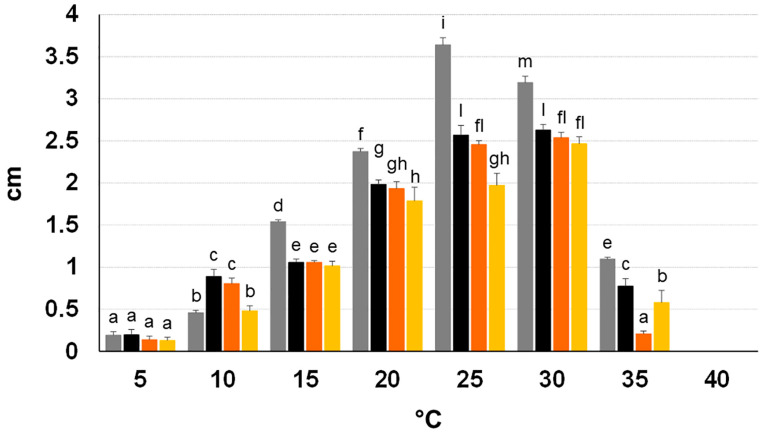
Daily growth rates of the *Diplodia seriata*-like isolates under study (CREA-DC TPR OL.437 in grey, CREA-DC TPR OL.464 in black, CREA-DC TPR OL.548 in orange, and CREA-DC TPR OL.700 in yellow) on PDA at different temperatures. Different letters indicate statistically significant differences (*p* < 0.05). The bars indicate the standard deviation of the mean. At 40 °C no growth occurred, but mycelium of all isolates was still viable after a six-day incubation.

**Figure 6 plants-13-02245-f006:**
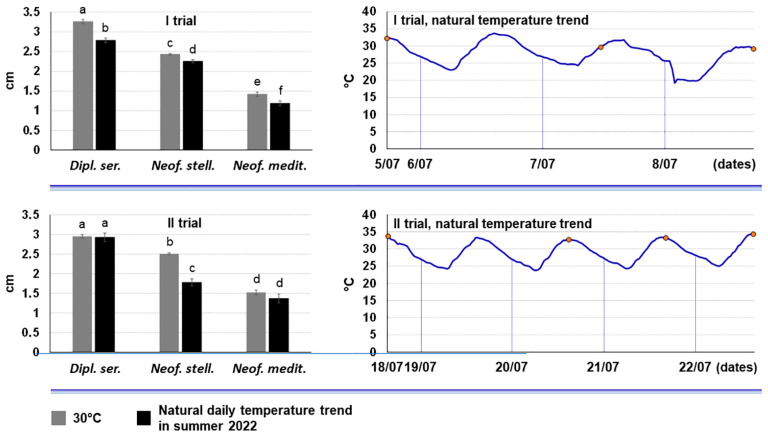
Growth rate of a *Diplodia seriata*-like isolate (CREA-DC TPR OL.437), *Neofusicoccum stellenboschiana* (CREA-DC TPR OL.431) and *N. mediterraneum* (CREA-DC TPR OL.427) at 30 °C (fast-growing temperature in vitro) compared with growth rate of the same fungi under the natural temperature trend occurring in July 2022. The bars indicate the standard deviation of the mean. Different letters indicate statistically significant differences (*p* < 0.05). In the first trial the I orange dot indicates the beginning of the trial, the II dot the final recording of *D. seriata*-like and the III dot refers to the recording of both *N. stellenboschiana* and *N. mediterraneum*; in the second trial, the I orange dot indicates the beginning of the trial, the II, III and IV dots refer to the final recording, respectively, of *D. seriata*-like, *N. stellenboschiana*, and *N. mediterraneum* (duration of the test was different for the different species due to their different growth rates).

**Figure 7 plants-13-02245-f007:**
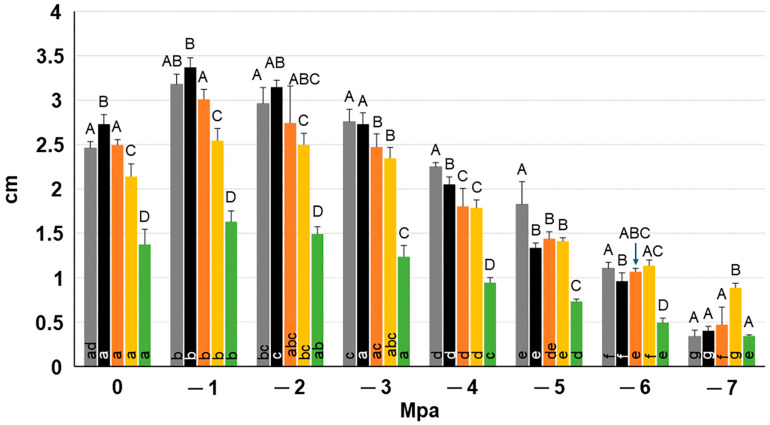
Daily growth rates of the *Diplodia seriata*-like isolates under study (CREA-DC TPR OL.437 in grey, CREA-DC TPR OL.464 in black, CREA-DC TPR OL.548 in orange, and CREA-DC TPR OL.700 in yellow) and *N. mediterraneum* (CREA-DC TPR OL.427) included for comparison (in green) on PDB amended with technical agar and different KCl doses to get different levels of water potential (Ψ measured in Mpa), at 25 °C. Different letters indicate statistically significant differences (*p* < 0.05). Uppercase letters at the top of the columns regard the comparison among the isolates for each value of Ψ. Lowercase letters at the base of the columns regard the comparison among the different Ψ for each isolate. The bars indicate the standard deviation of the mean.

**Figure 8 plants-13-02245-f008:**
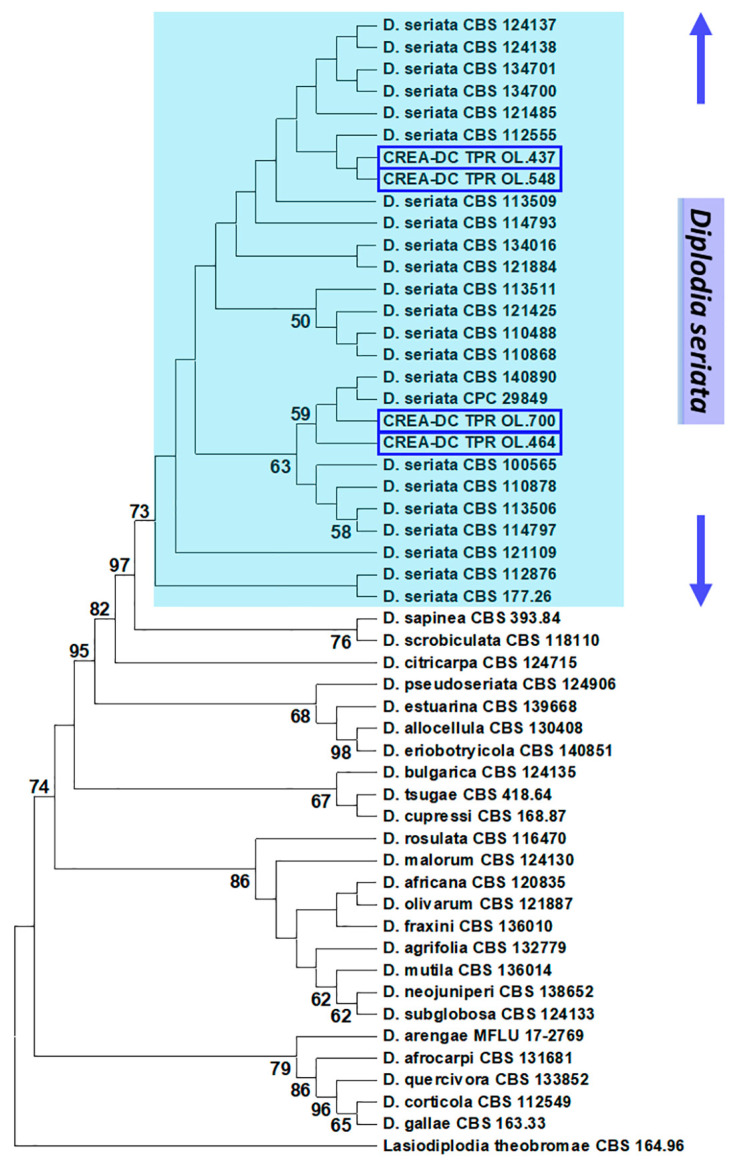
Phylogenetic tree of *Diplodia* species, based on ITS + TEF1-α + TUB2 data set and including the botryosphaeriaceous isolates from Salento (Apulia, Italy): CREA-DC TPR OL.437, 464, 548, 700. *Diplodia seriata* clade is shaded in blue-sky. The evolutionary history was inferred by using the Maximum Likelihood method and General Time Reversible model [45]. The tree with the highest log likelihood (−4846.06) is shown. The percentage of trees in which the associated taxa clustered together (bootstrap support value) is shown next to the branches only for values higher than 49%. The tree is drawn to scale, with branch lengths measured in the number of substitutions per site. This analysis involved 52 nucleotide sequences. There were a total of 1281 positions in the final dataset. Evolutionary analyses were conducted in MEGA X [46]. The arrows and the azure-shaded area highlight the *Diplodia seriata* clade. The boxes highlight the *Diplodia* isolates under study.

**Figure 9 plants-13-02245-f009:**
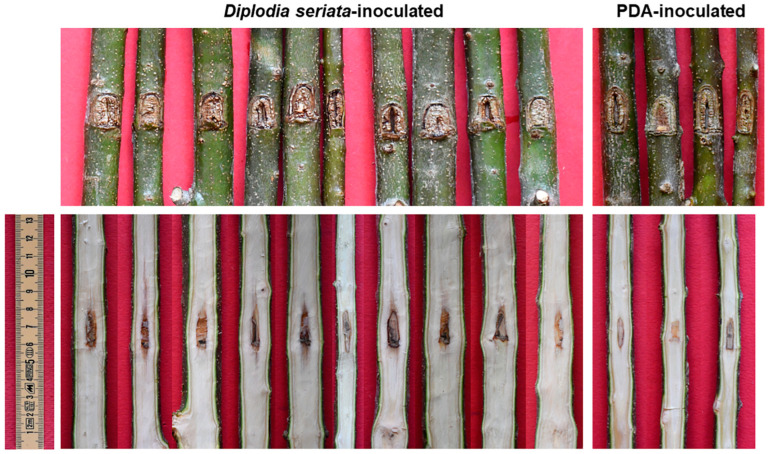
Outcome of inoculations with *Diplodia seriata* (CREA-DC TPR OL.437) on the stem of olive trees, cv Frantoio, in terms of bark reactions and wood discolouration (all replicates are shown). Test duration: eleven months. Note that the inoculation points healed similarly to controls and a restricted, pale, and thread-like discolouration was present in the wood above and below the inoculation point.

**Figure 10 plants-13-02245-f010:**
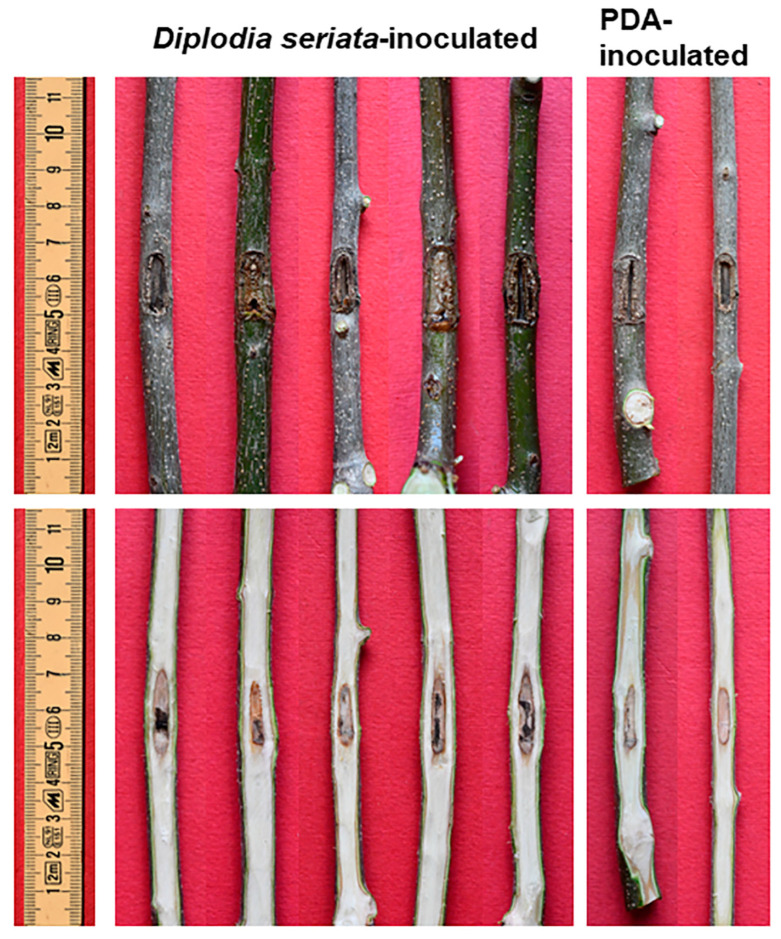
Outcome of inoculations with *Diplodia seriata* (CREA-DC TPR OL.437) on one-year-old twigs of olive trees, cv Frantoio, in terms of bark reactions and wood discolouration (some representative replicates are shown). Test duration: eleven months. Note that the inoculation points healed similarly to controls and a restricted, pale, and thread-like discolouration was present in the wood above and below the inoculation point.

**Figure 11 plants-13-02245-f011:**
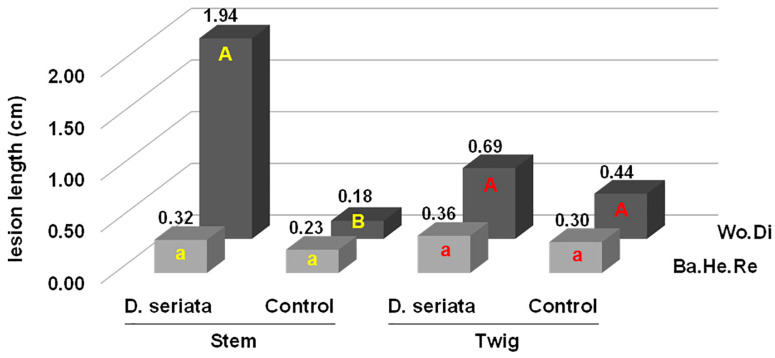
Graphic representation of the outcome of inoculations with *Diplodia seriata* (CREA-DC TPR OL.437) on stems and twigs of olive trees, cv Frantoio (Figure 9 and Figure 10). The length of the external healing reactions and of wood discolouration streaks are the net of the native inoculation wound. Ba.He.Re. = bark healing reactions, Wo.Di. = wood discolouration. Statistical analyses were conducted separately for bark (lower case letters) and wood (capital letters) and for stem (yellow letters) and twigs (red letters). Different letters indicate statistically significant differences (*p* < 0.01).

**Figure 12 plants-13-02245-f012:**
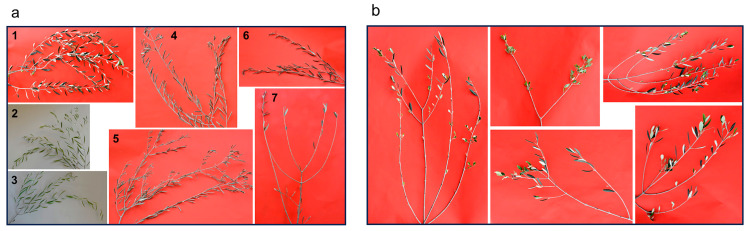
The different phenotypic phases of olive tree subjected to a prolonged water lack. (**a**) Under the effect of water stress, leaf blades progressively rolled downward. The numbered pictures represent the different phases: phase 1 is the condition of a well-watered trees, while in the subsequent pictures (from 2 to 7) the effect of increasing water stress is represented. When 80–100% of the leaves were rolled but still flexible, trees were inoculated with *Diplodia seriata* (CREA-DC TPR OL.437) and watering was progressively resumed (see the text) (phase 3–4). After the inoculation, over time, though watering had been resumed, most leaves desiccated and dropped (phases from 5 to 7). Twigs preserved their viability and only distal ends might be desiccated. (**b**) Resprouting followed leaf fall thus showing that the degree of water stress that we imposed on the trees was not lethal and did not compromise twig viability.

**Figure 13 plants-13-02245-f013:**
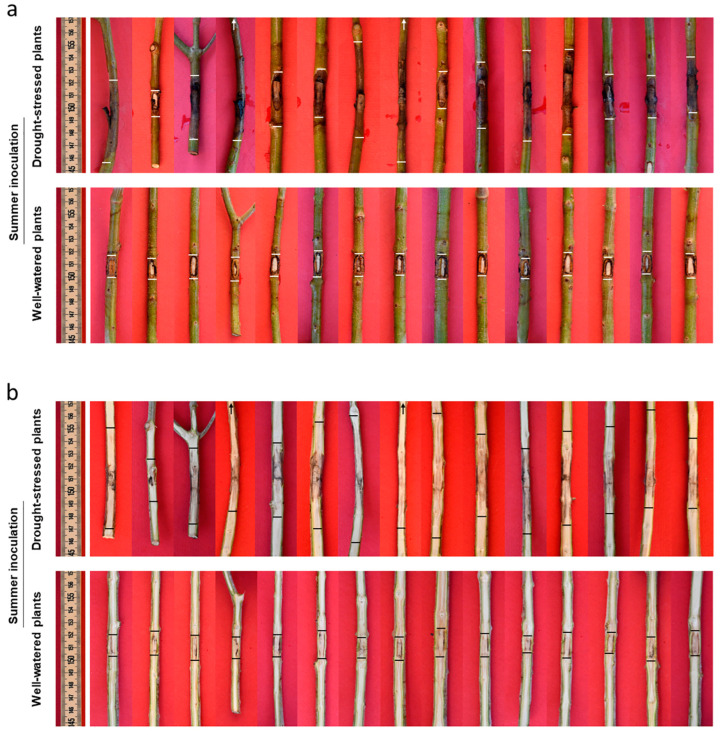
Outcome of inoculations with *Diplodia seriata* (CREA-DC TPR OL.437) of drought-stressed (DS) olive trees compared with those well-watered (WW). The test was conducted in summer over 35 days. (**a**) Bark necrosis in DS trees and bark healing in WW trees; (**b**) wood discolouration in DS and WW trees. The dashes define the lower and the upper limits of the lesions. The arrows indicate a spread of the necrosis beyond the limit of the figure.

**Figure 14 plants-13-02245-f014:**
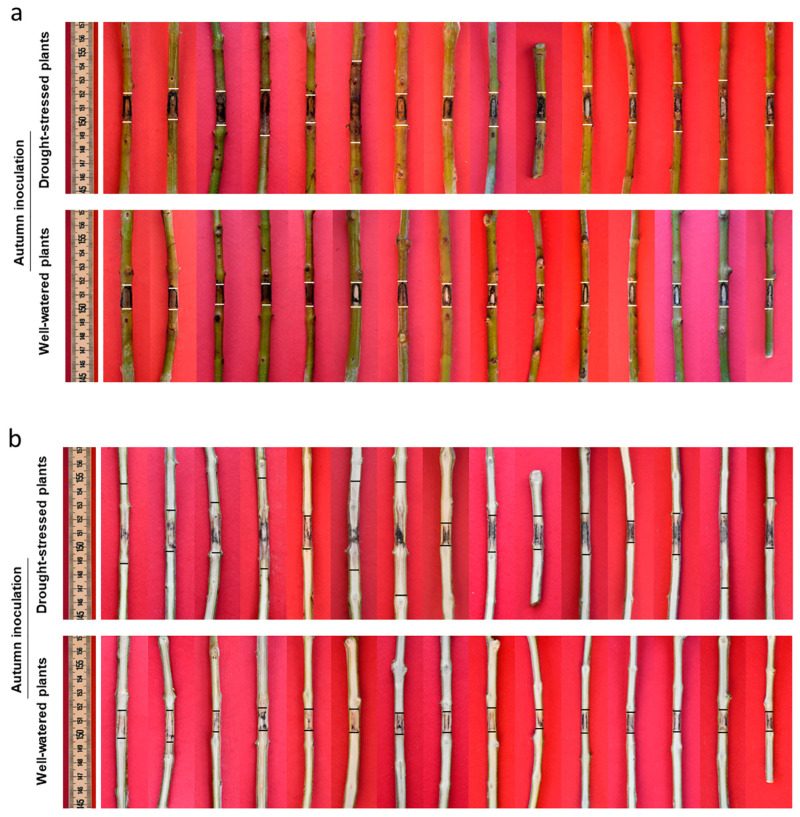
Outcome of inoculations with *Diplodia seriata* (CREA-DC TPR OL. 437) of drought-stressed (DS) olive trees compared with those well-watered (WW). The test was conducted in autumn over 35 days. (**a**) Bark necrosis in DS trees and bark healing in WW trees; (**b**) wood discolouration in DS and WW trees. The dashes define the lower and the upper limits of the lesions.

**Figure 15 plants-13-02245-f015:**
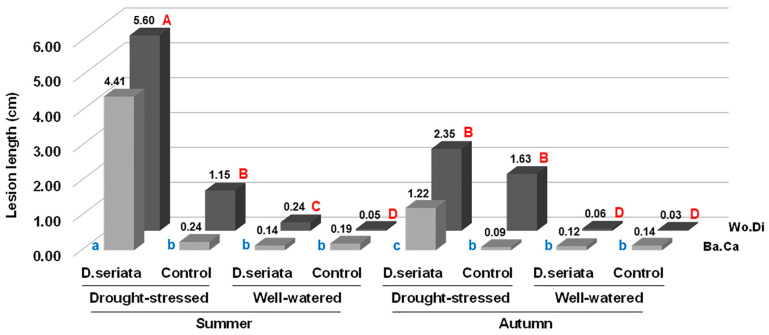
Graphic representation of the outcome of inoculations conducted in summer and in autumn with *Diplodia seriata* (CREA-DC TPR OL.437) (Figure 13 and Figure 14) on one-year-old twigs of drought-stressed (DS) olive trees, and compared with those well-watered (WW). The length of the external bark canker/healing reactions and of wood discolouration streaks are the net of the native inoculation wound. Externally, only *D. seriata*/DS/summer and *D. seriata*/DS/autumn showed bark canker; in the other treatments, the columns refer to bark healing reactions. Ba.Ca. = bark canker, Wo.Di. = wood discolouration. Statistical analyses were conducted separately for bark (lower case letters in azure) and wood (capital letters in red) and comparing all treatments. Different letters indicate statistically significant differences (*p* < 0.01).

**Figure 16 plants-13-02245-f016:**
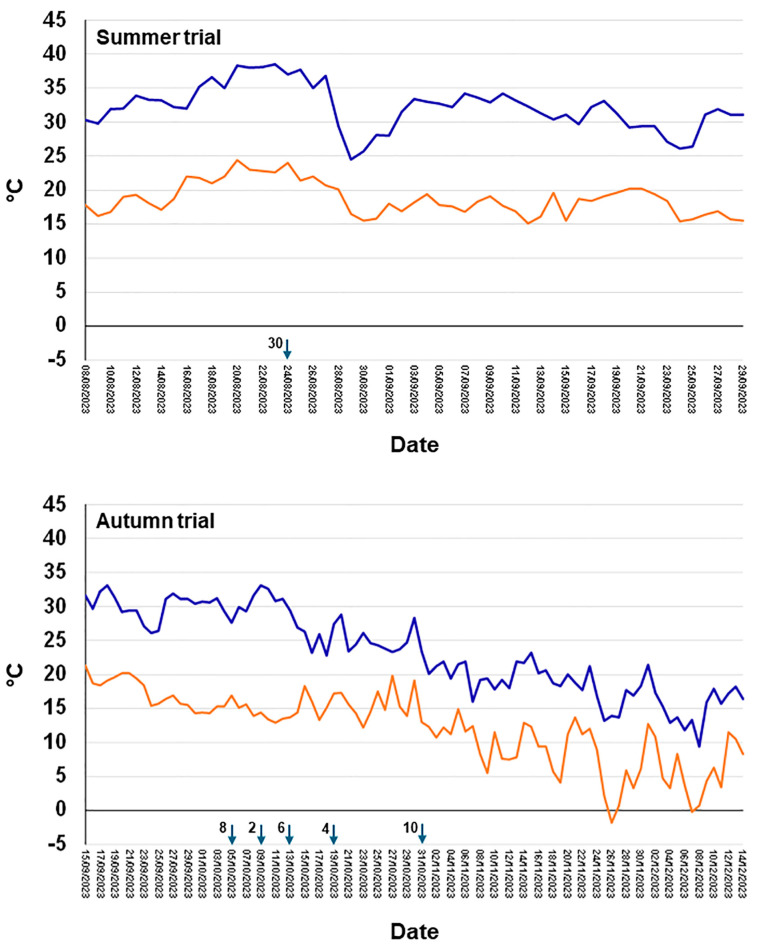
Minimum (curve in orange) and maximum (curve in blue) daily temperature trend occurring during pathogenicity trials conducted in summer and in autumn with *Diplodia seriata* (CREA-DC TPR OL.437) on one-year-old twigs of olive trees subjected to drought stress (DS) and compared with those well-watered (WW). Temperature recording was from the beginning of the interruption of water supply till the end of the trials when inoculated twigs were harvested and symptoms recorded. The arrows indicated the inoculation day(s) and the numbers beside them are the trees that were inoculated (half were DS, the other half were WW). Note that in the autumn trial inoculations were performed in a staggered order as soon as the trees showed the inoculation phenotype.

**Figure 17 plants-13-02245-f017:**
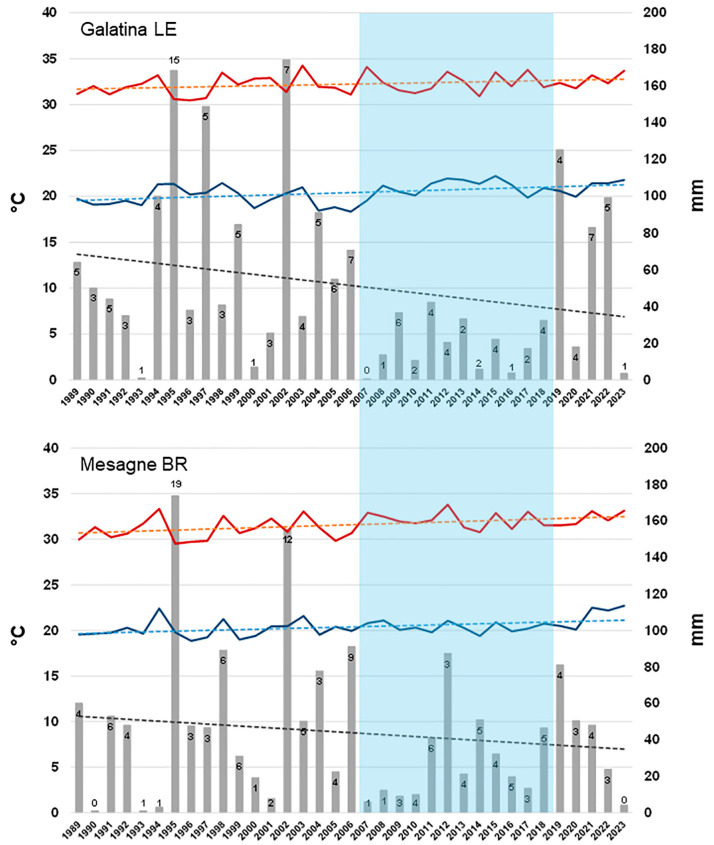
Meteorological trends, temperature and rainfall, over the last 35 years (1989–2023) in summer (July and August) in two different locations in Salento (Apulia, Italy). For each year they represent the average maximum (continuous red line) and minimum (continuous blue line) temperatures of the period July–August, as well as the total amount of the rainfall that occurred in the same period (grey bars, with numerical values representing the number of rainy days). The dashed lines are their trend lines. The azure-shaded area evidences the period of time (2007–2018) in which rainfall was constantly scarce in July–August, thus defining a prolonged period of drought.

**Figure 18 plants-13-02245-f018:**
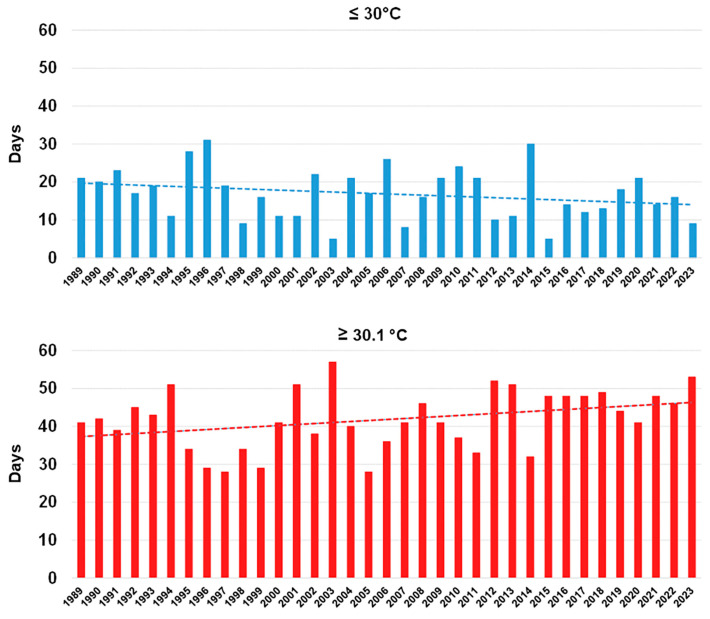
Number of days in which maximum temperatures reached values ≤ 30 °C and ≥30.1 °C in the period July–August in Galatina (LE) and in the time span 1989–2023. Dashed lines are the trend lines.

**Table 1 plants-13-02245-t001:** Conidia size of the *Diplodia seriata*-like isolates from olive trees of Salento (Apulia, Italy). Fifty conidia were measured for each isolate.

*Diplodia seriata*-like Isolates	Length Mean (Range, SD) (µm)	Width Mean (Range, SD) (µm)	Length/Width
CREA-DC TPR OL.437 (Mesagne, Brindisi)	24.9 (20.4–28.0, 1.3)	12.1 (10.5–13.2, 0.6)	2.1
CREA-DC TPR OL.464 (Nardò, Lecce)	26.2 (22.9–34.7, 2.5)	11.5 (9.9–16.1, 1.1)	2.3
CREA-DC TPR OL.548 (Sava, Taranto)	25.2 (21.3–31.5, 2.0)	10.8 (9.1–13.3, 0.7)	2.3
CREA-DC TPR OL.700 (Tricase, Lecce)	26.8 (23.6–30.8, 1.7)	11.0 (10.1–12.0, 0.5)	2.4

**Table 2 plants-13-02245-t002:** Survival degree of the *Diplodia seriata*-like mycelial plugs to a 6 h heat treatment.

*Diplodia seriata* Isolate	45 °C (%)	50 °C (%)	55 °C (%)
CREA-DC TPR OL.437	15/15 (100)	7/15 (46.7)	0/15 (0)
CREA-DC TPR OL.464	15/15 (100)	6/15 (40)	1/15 (6.7)
CREA-DC TPR OL.548	15/15 (100)	5/15 (33.3)	2/15 (13.3)
CREA-DC TPR OL.700	15/15 (100)	4/15 (26.7)	0/15 (0)

**Table 3 plants-13-02245-t003:** Inoculation trials performed on olive tree with *Diplodia seriata* on the main stem and one-year-old twigs of well-trained non-stressed trees (basic pathogenicity trials) and drought-stressed trees inoculated in full summer and at the beginning of the autumn compared with non-stressed controls (water-conditioned pathogenicity trials).

Basic Pathogenicity Trials	Age of the Trees (Year)	Cultivar	Biological Replicates	Inoculation Time	Duration *	Average Ø at the Inoculation Point (cm)
Stem	2/3	Frantoio	10	27 June 2022	11 mo	1.2
Twig	2/3	Frantoio	20	27 June 2022	11 mo	0.7
Twig	7/8	Ogliarola	20	3 October 2023	5 mo	0.7
**Water-conditioned pathogenicity trials ****						
Twig (non-str)	2/3	Frantoio	15	24 August 2023	35 ds	0.8
Twig (str)	2/3	Frantoio	15	24 August 2023	35 ds	0.8
Twig (non-str)	2/3	Frantoio	15	5–31 October 2023	35–55 ds	0.7
Twig (str)	2/3	Frantoio	15	5–31 October 2023	35–55 ds	0.7

* mo = month, ds = days. ** non-str = non-stressed, str = stressed.

## Data Availability

Data are contained within the article and Appendix A.

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
