# Peer review of "Diplodia seriata Isolated from Declining Olive Trees in Salento (Apulia, Italy): Pathogenicity Trials Give a Glimpse That It Is More Virulent to Drought-Stressed Olive Trees and in a Warmth-Conditioned Environment"

_plants, 2024, doi:10.3390/plants13162245_

Round 1

Reviewer 1 Report

Comments and Suggestions for Authors

In this paper, the authors report the finding in Salento of Diplodia seriata, another Botryosphaeriaceae species, in Xylella fastidiosa-infected olive trees affected by symptoms of branch except Neofusicoccum mediterraneum and N. stellenboschiana in olive. They provided systemically and complete information about this agent in In Salento (Apulia, Italy) through 1989-2023. Results of this paper showed heat and drought act as predisposing/inciting factors facilitating D. seriata as a contributor. The fact that several adverse factors, biotic and abiotic, are simultaneously burdening olive trees in Salento.

Author Response

Dear reviewer

Thanks a lot for the appreciation of our work.

Comments and Suggestions for Authors

In this paper, the authors report the finding in Salento of Diplodia seriata, another Botryosphaeriaceae species, in Xylella fastidiosa-infected olive trees affected by symptoms of branch except Neofusicoccum mediterraneum and N. stellenboschiana in olive. They provided systemically and complete information about this agent in In Salento (Apulia, Italy) through 1989-2023. Results of this paper showed heat and drought act as predisposing/inciting factors facilitating D. seriata as a contributor. The fact that several adverse factors, biotic and abiotic, are simultaneously burdening olive trees in Salento.

Best regards

Massimo Pilotti

Reviewer 2 Report

Comments and Suggestions for Authors

The manuscript of Manetti et al., entitled “Diplodia seriata isolated from declining olive trees in Salento (Apulia, Italy): pathogenicity trials give a glimpse that it is more virulent to drought-stressed olive trees and in a warmth-conditioned environment” describes the analysis of Diplodia seriata species for their contribution in olive decline in Salento. The strains, all derived from Xylella infested olive trees, were morphologically and pathogenetically tested under various conditions. This reveals that the D. seriate species can contribute to olive decline in the presence of predisposing/inciting factors. The manuscripts has several issues that need to be addressed. Grammatically, the manuscript is quite poor, there are many sentences that can be improved (interpunction, missing/extra words in the text, word order etc.., some examples given below) which will contribute to a more smooth reading of the text, understanding what is really meant, and potentially reduces the amount of text. This in combination of restructuring the introduction, transfer/delete some text from the review and discussion section should contribute to a more concise manuscript. Potentially, a native writer should be contacted who also keeps eye on the goal of the manuscript. The next issue is the choice of the “pathogens”, notably, they are all obtained from Xylella infected trees. It would be valuable when the >90% of OQDS symptomatic plants would provide the same strains and thus the connection between plant symptoms and disease are more elucidated. The pathogenicity test, using a massive infection on cut material is maybe essential but under pressure (drought/primary infections) many pathogen related endophytes show similar behavior. Notably, the authors lack to indicate some aspects such as potential genetic differences that are aiding in a pathosystem. In different fungal plant diseases the role of specific effector proteins, often genetically located on extra chromosomes, allowing rapid evolution, are important (fusarium/Phytophthora etc). In this study, with only the acknowledgement that experimental conditions are difficult to compare based on host plant, fungal strain/isolates, inoculation methods etc…in order to describe differences in virulence, this might also play a role. Since no genetic analysis was performed, only species level identification on household genes, this needs to be indicated. This might explains the pathogenic behavior difference between various strains.

Some (but not all) of the points related to grammatic issues to address:

Line 50-53, please rewrite sentence.

Line 54, should read: among which are

Line 58, remove additional

Line 62, remove even, name an example

Line 71, the last of the sentence is not correct

Line 77-79, place elsewhere more in line with the flow of the introduction

Line 86, this sentence should be rewritten

Line 93, again sentence rewrite to get it more smoothly

Line 104, place elsewhere in flow of introduction and not as separate paragraph

Line 106, headline should be corrected, either olive treeS or something else

Line 112, the part is on Italy, what’s the need to mention again?

Line 114, no need to give details on the pathogenicity without any other comparison

Line 180, is this agar or pine needles itself (see Figure 2)

Line 217, “the PDA test” or a PDA growth test.

Line 224, … maintained viability in all the plugs, no need for the

Line 292-293, should benefit from a comma after “tree.

Line 297, consistency for Figure (with or without capital).

Figure 12, please remake the figure with proper alignments, now it looks very messy

Line 394, rewrite sentence

Line 504, rainfall decrease of 18.5 mm or rainfall decreased with 18.5 mm

Names of strains, 1e time always in full, in the discussion part there are several introduced with the abbreviated name.

Review and Discussion is way too long. Assuming that the authors want to be complete, the trend is mostly clear without all details. Also, when discussing results from this study, no need to repeat details (e.g. specific values).

Line 852-857… this is one sentence, poor interpunction, not clear what is the main message.

Comments on the Quality of English Language

See Comments

Author Response

Dear reviewer

Thanks a lot for your work.

See below a point by point replay to your comments.

Reviewer II

The manuscript of Manetti et al., entitled “Diplodia seriata isolated from declining olive trees in Salento (Apulia, Italy): pathogenicity trials give a glimpse that it is more virulent to drought-stressed olive trees and in a warmth-conditioned environment” describes the analysis of Diplodia seriata species for their contribution in olive decline in Salento. The strains, all derived from Xylella infested olive trees, were morphologically and pathogenetically tested under various conditions. This reveals that the D. seriate species can contribute to olive decline in the presence of predisposing/inciting factors. The manuscripts has several issues that need to be addressed. Grammatically, the manuscript is quite poor, there are many sentences that can be improved (interpunction, missing/extra words in the text, word order etc.., some examples given below) which will contribute to a more smooth reading of the text, understanding what is really meant, and potentially reduces the amount of text.

  1. Pilotti

Yes, we revised the English language of the proof

Reviewer II

This in combination of restructuring the introduction, transfer/delete some text from the review and discussion section should contribute to a more concise manuscript. Potentially, a native writer should be contacted who also keeps eye on the goal of the manuscript. The next issue is the choice of the “pathogens”, notably, they are all obtained from Xylella infected trees. It would be valuable when the >90% of OQDS symptomatic plants would provide the same strains and thus the connection between plant symptoms and disease are more elucidated.

  1. Pilotti

Yes, that is right. In fact, we are conducting a survey in the territory of Salento and are collecting several samples from declining olive trees. Samples are being analysed with isolation on artificial media and isolated fungi/Botryosphaeriaceae are being identified with multilocus-sequencing. Contemporary a metagenomic approach is also in progress. This will address the crucial issue of the association between symptom, Xylella fastidosa and Botryosphaeriaceae. But all this will be the focus of a distinct scientific article, probably the next one. So, this is not the object of this paper which instead want to address the characterization of the virulence of D. seriata under different conditions.

Reviewer II

The pathogenicity test, using a massive infection on cut material is maybe essential but under pressure (drought/primary infections) many pathogen related endophytes show similar behavior.

M.Pilotti

If in the olive many endophytes show a similar behavior has to be demonstrated. In my knowledge these type of study are not common. Nevertheless, as scientists, we have not to undervalue those microorganisms that are considered avirulent, but they reveal a significant virulence degree when they are “facilitated” by other abiotic exogenous factors.

Reviewer II

Notably, the authors lack to indicate some aspects such as potential genetic differences that are aiding in a pathosystem. In different fungal plant diseases the role of specific effector proteins, often genetically located on extra chromosomes, allowing rapid evolution, are important (fusarium/Phytophthora etc). In this study, with only the acknowledgement that experimental conditions are difficult to compare based on host plant, fungal strain/isolates, inoculation methods etc…in order to describe differences in virulence, this might also play a role. Since no genetic analysis was performed, only species level identification on household genes, this needs to be indicated. This might explains the pathogenic behavior difference between various strains.

  1. Pilotti

That genetic differences among the isolates of D. seriata might account for diversity in virulence is discussed and supported in the manuscript (see lines 689-691, 709-718). However, we added an additional part to be more explicit (lines 728-734).

Some (but not all) of the points related to grammatic issues to address:

Line 50-53, please rewrite sentence.

  1. Pilotti

OK I simplified the sentence

Line 54, should read: among which are

  1. Pilotti

OK I modified

Line 58, remove additional

  1. Pilotti

OK I did it

Line 62, remove even, name an example

  1. Pilotti

OK I did it

Line 71, the last of the sentence is not correct

  1. Pilotti

I simplified the sentence

Line 77-79, place elsewhere more in line with the flow of the introduction

  1. Pilotti

I think that this part is located appropriately as it is included soon before the part dedicated to the olive tree. So, if this issue is not decisive for you, I would prefer not to move elsewhere this part.

Line 86, this sentence should be rewritten

  1. Pilotti

OK, I rephrased

Line 93, again sentence rewrite to get it more smoothly

  1. Pilotti

OK, I rephrased

Line 104, place elsewhere in flow of introduction and not as separate paragraph

  1. Pilotti

OK I relocated the sentence

Line 106, headline should be corrected, either olive treeS or something else

  1. Pilotti

I rephrased: “Botryosphaeriaceae are emerging in olive tree in Italy”

Line 112, the part is on Italy, what’s the need to mention again?

  1. Pilotti

I rephrased: “Linaldeddu et al. [34] found also additional Botryosphaeriaceae species in Veneto, Lombardy, Sardinia and Calabria, including D. seriata….”

Line 114, no need to give details on the pathogenicity without any other comparison

M.Pilotti

I rephrased: “…this fungus produced very small lesions four months after the inoculation….

Line 180, is this agar or pine needles itself (see Figure 2)

  1. Pilotti

Pine needles deposed on agar plates

Line 217, “the PDA test” or a PDA growth test.

Line 224, … maintained viability in all the plugs, no need for the

  1. Pilotti

 “the” has just been removed by the Journal, I suppose

Line 292-293, should benefit from a comma after “tree.

  1. Pilotti

I’ve added

Line 297, consistency for Figure (with or without capital).

  1. Pilotti

I’ve corrected

Figure 12, please remake the figure with proper alignments, now it looks very messy

  1. Pilotti

I modified the figure realigning the photos 

Line 394, rewrite sentence

  1. Pilotti

I don’t understand how I would have to change

Line 504, rainfall decrease of 18.5 mm or rainfall decreased with 18.5 mm

  1. Pilotti

I don’t understand how I would have to change

Names of strains, 1e time always in full, in the discussion part there are several introduced with the abbreviated name.

  1. Pilotti

I checked the manuscript. More like I found cases of species cited in full though they had been cited previously (I corrected)

Review and Discussion is way too long. Assuming that the authors want to be complete, the trend is mostly clear without all details. Also, when discussing results from this study, no need to repeat details (e.g. specific values).

M.Pilotti

Indeed, we cite details of results in the discussion to compare them with corresponding details from literature, which would give strength to our results

Line 852-857… this is one sentence, poor interpunction, not clear what is the main message.

  1. Pilotti

Done. I hope it is appropriate

Dear Reviewer,

note that all modifications that you proposed and that we made, are easily recognizable in the text as they are marked with a comment that is addressed to Reviewer II. However, due to extensive modification of the text, the original numbering of lines (and that you indicate in the revision) has been changed.

Best regards

Massimo Pilotti

Reviewer 3 Report

Comments and Suggestions for Authors

This manuscript isolated a fungal species of Botryosphaeriaceae, Diplodia seriata, from declining olive trees in Salento (Apulia, Italy). They identified this species based on morphological features and a multilocus phylogeny, and detected the pathogenicity under diverse coniditions. They concluded that heat and drought act as predisposing/inciting factors facilitating D. seriata as a contributor.

This manuscript is very interesting, and the findings may help to control of olive declines. Whereas the manuscript is very long and wordy, the authors needs to make appropriate simplifications. 

The keywords are too much.

Line 118, "Olive Quick Decline Syndrome (OQDS)", a comma is missed after this sentnece.

Line 138, "[31,32]", a comma is missed after this sentnece.

Line 141, "In this work", a comma is needed here.

Line 163, "BTD-affected olive trees", a comma is needed here.

Line 526-527,According to Batista et al. [1] D. seriata has been reported on 121 hosts by genus and globally distributed in 46 countries” changed to "According to previous studies, D. seriata has been reported on 121 hosts by genus and globally distributed in 46 countries [1] "

Author Response

Dear reviewer

Thanks a lot for the appreciation of our work. We made all suggested changes

See below a point by point replay.

Comments and Suggestions for Authors

This manuscript isolated a fungal species of Botryosphaeriaceae, Diplodia seriata, from declining olive trees in Salento (Apulia, Italy). They identified this species based on morphological features and a multilocus phylogeny, and detected the pathogenicity under diverse coniditions. They concluded that heat and drought act as predisposing/inciting factors facilitating D. seriata as a contributor.

This manuscript is very interesting, and the findings may help to control of olive declines. Whereas the manuscript is very long and wordy, the authors needs to make appropriate simplifications. 

The keywords are too much.

  1. Pilotti

I removed “Olive Quick Decline Syndrome OQDS”, “Temperature” and “Rainfall”.

Line 118, "Olive Quick Decline Syndrome (OQDS)", a comma is missed after this sentnece.

Line 138, "[31,32]", a comma is missed after this sentnece.

Line 141, "In this work", a comma is needed here.

Line 163, "BTD-affected olive trees", a comma is needed here.

Line 526-527,“According to Batista et al. [1] D. seriata has been reported on 121 hosts by genus and globally distributed in 46 countries” changed to "According to previous studies, D. seriata has been reported on 121 hosts by genus and globally distributed in 46 countries [1] "

  1. Pilotti

I made all the above suggested changes

Dear Reviewer,

note that all modifications that you proposed and that we made, are easily recognizable in the text as they are marked with a comment that is addressed to Reviewer III. However, due to extensive modification of the text, the original numbering of lines (that you indicated in your revision) has been changed.

Best regards

Massimo pilotti

Round 2

Reviewer 2 Report

Comments and Suggestions for Authors

Most of my points were addressed/clarified to my satisfaction. Still feel that grammatically improvements could be made, which I leave to the Journal to judge. Some obvious points remain to be corrected (see below) at ;east. Note that after accepting the changes, one should scrutinize the manuscript for unintended and overlooked mistakes (e.g. spacing and extra words).

Line 92-93: use “were identified as” instead of resulted; agents instead of players

Line 99: please check for capital with Botryosphaeriaceous

Line 105: remove an

Line 144 (145): … a morphotype is emerging as widespread that was identified as D. seriata.

Line 148: we consider it urgent..

Line 211: add a . after new sentence

Line 234: the Figure 5 does show the absence of growth but not the viability (Ill placed Figure indication). On top of that, knowing that conidia and other structures are produced by these fungi, what is the evidence that mycelium maintained the viability?

Line 263: in vitro should be in italics

Line 671: as indicated before, details can be too much: e.g. here it should be enough to say: D. seriata was among the five most virulent species. (all the rest is not important and 54 mm is not informative without additional information). In this paragraph many more details might be removed..

Line 680, only one tree?

Line 684: suggestion to change into: from Croatia were ranked as highly virulent

Line 691: thanks for indicating but my notes were on the fungal site. When then pointing out the effect as dependence on plant phenological stage rather than a comparison of fungal isolates on a range of plant varieties this is a bit weak. Btw: should read instance

Line 1088: supplemented (typo)

Comments on the Quality of English Language

see general comments

Author Response

Dear Reviewer

thanks for this additional round of corrections and comments

See below my point by point replay

Best regards

Massimo Pilotti

Most of my points were addressed/clarified to my satisfaction. Still feel that grammatically improvements could be made, which I leave to the Journal to judge. Some obvious points remain to be corrected (see below) at ;east. Note that after accepting the changes, one should scrutinize the manuscript for unintended and overlooked mistakes (e.g. spacing and extra words).

Line 92-93: use “were identified as” instead of resulted; agents instead of players

M.Pilotti

OK see line 89

Line 99: please check for capital with Botryosphaeriaceous

  1. Pilotti

I think that botryosphaeriaceous is an adjective, so why use the capital?....

Line 105: remove an

M.Pilotti

OK see line 100

Line 144 (145): … a morphotype is emerging as widespread that was identified as D. seriata.

M.Pilotti

OK see line 135

Line 148: we consider it urgent..

M.Pilotti

OK see line 138

Line 211: add a . after new sentence

M.Pilotti

Sorry I don’t understand

Line 234: the Figure 5 does show the absence of growth but not the viability (Ill placed Figure indication). On top of that, knowing that conidia and other structures are produced by these fungi, what is the evidence that mycelium maintained the viability?

  1. Pilotti

the plates that had been incubated at 40°C for six days, were transferred at 25°C and growth started demonstrating viability of the mycelium (described in “Materials and methods, lines 924-925)

Line 263: in vitro should be in italics

M.Pilotti

OK see line 252

Line 671: as indicated before, details can be too much: e.g. here it should be enough to say: D. seriata was among the five most virulent species. (all the rest is not important and 54 mm is not informative without additional information). In this paragraph many more details might be removed..

M.Pilotti

I revised this part and eliminated some details. In general, I would wish that this part report some details of pathogenicity trials carried out by the different authors, in the different part of the world. Though the different experiments are not strictly comparable, the whole picture helps to evaluate the virulence degree of D. seriata compared with the other Botryosphaeriaceae species.

Line 680, only one tree?

M.Pilotti

OK see line 653

Line 684: suggestion to change into: from Croatia were ranked as highly virulent

M.Pilotti

OK see line 657

Line 691: thanks for indicating but my notes were on the fungal site. When then pointing out the effect as dependence on plant phenological stage rather than a comparison of fungal isolates on a range of plant varieties this is a bit weak. Btw: should read instance

Line 1088: supplemented (typo)

M.Pilotti

OK see line 1054